# DICEPTION: A Generalist Diffusion Model for Visual Perceptual Tasks

**Canyu Zhao**[1]   **Yanlong Sun**[2]   **Mingyu Liu**[1]   **Huanyi Zheng**[1]   **Muzhi Zhu**[1]
**Zhiyue Zhao**[1]   **Hao Chen**[1]   **Tong He**[1,3]   **Chunhua Shen**[1,4,*]

[1] Zhejiang University    [2] Tsinghua University
[3] Shanghai AI Laboratory    [4] Zhejiang University of Technology

## Abstract

This paper's primary objective is to develop a robust generalist perception model capable of addressing multiple tasks under constraints of computational resources and limited training data. We leverage text-to-image diffusion models pre-trained on billions of images and successfully introduce our DICEPTION, a visual generalist model. Exhaustive evaluations demonstrate that DICEPTION effectively tackles diverse perception tasks, even achieving performance comparable to SOTA single-task specialist models. Specifically, **we achieve results on par with SAM-vit-h using only 0.06% of their data (*e.g.*, 600K vs. 1B pixel-level annotated images)**. We designed comprehensive experiments on architectures and input paradigms, demonstrating that the key to successfully re-purposing a single diffusion model for multiple perception tasks lies in maximizing the preservation of the pre-trained model's prior knowledge. Consequently, DICEPTION can be trained with substantially lower computational costs than conventional models requiring training from scratch. Furthermore, adapting DICEPTION to novel tasks is highly efficient, necessitating fine-tuning on as few as 50 images and approximately 1% of its parameters. Finally, we demonstrate that a subtle application of classifier-free guidance can improve the model's performance on depth and normal estimation. We also show that pixel-aligned training, as is characteristic of perception tasks, significantly enhances the model's ability to preserve fine details. DICEPTION offers valuable insights and presents a promising direction for the development of advanced diffusion-based visual generalist models.

## 1 Introduction

Foundation models [51, 90, 125, 126, 123, 11, 7, 86, 78, 94, 6, 40], typically requiring extensive training on billions of data samples, play a pivotal role in their respective domains. In natural language processing (NLP), current foundation models [9, 105, 106, 27] have already demonstrated the potential to serve as versatile solutions, solving diverse fundamental tasks and with minimal fine-tuning needed for new tasks. This success can be attributed to the relatively small representational differences among various language tasks. However, in the domain of computer vision, task representations can differ substantially, and up to date, we still lack an effective approach to unify these distinct tasks. Consequently, existing vision foundation models usually excel at one single specific task, such as image segmentation [51, 90] or monocular depth estimation [125, 126, 123], because they are trained on data tailored exclusively to that task. Owing to the pronounced disparity in visual representations across tasks, coupled with the single-task specialization that characterizes current vision foundation models, fine-tuning these models for new tasks remains a formidable

---

*Corresponding author.

39th Conference on Neural Information Processing Systems (NeurIPS 2025).

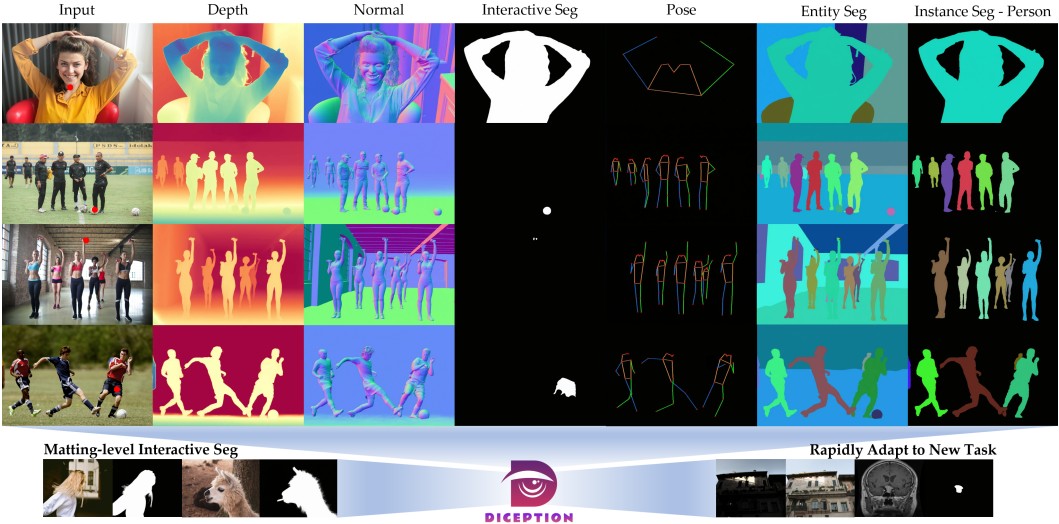

| Input | Depth | Normal | Interactive Seg | Pose | Entity Seg | Instance Seg - Person |

**Matting-level Interactive Seg**    **Rapidly Adapt to New Task**

DICEPTION

Figure 1: **With one single model**, DICEPTION solves 6 perception tasks without relying on any task-specific modules. The red dots in the figure indicate the input points used for interactive segmentation. DICEPTION can quickly adapt to new tasks by fine-tuning less than 1% of its parameters on as few as 50 images. **For additional visualizations, please refer to Figures S8, S11, S10, S15, S16, S17, S18, S19, S20, S21, S22 in the Appendix.** We select Person as the instance segmentation example for consistent visualization. Our method is limited to only human instances.

challenge. Despite efforts [12, 78, 40, 91] to learn universal visual representations, these models still falls noticeably short compared to specialized models in specific tasks.

Recent studies [115, 71, 70, 75, 2, 119] on visual generalist models are predominantly trained from scratch, often requiring substantial computational resources and large datasets to achieve good results. Unfortunately, the price of collecting a sufficiently large and high-quality multi-task dataset is substantial. Here, inspired by the success of diffusion models, we propose the hypothesis that leveraging their powerful priors can help mitigate the significant computational and data overhead for training powerful generalist models. While some existing works [49, 120, 39, 129, 96] have demonstrated that this is feasible in single-task scenarios, the potential of diffusion model priors in multi-task settings remains largely under-explored.

In this paper, we successfully leverage the priors of diffusion models to achieve results on par with the state-of-the-art models on various tasks with only minimal training data. We name our powerful visual generalist model **DICEPTION**. For each task, we require substantially less data than specialized foundation models. For instance, compared to SAM segmentation trained on 1 billion pixel-level annotated samples, DICEPTION *achieves comparable performance using a significantly smaller dataset of 600K samples*, without any training data cherry-picking.

More significantly, DICEPTION highlights that *the generative image priors lead to surprisingly more efficient and effective pathways to generalist image understanding models.* We analyze a series of design choices for transferring one single modern diffusion model to multiple perception tasks, and identify that the key to successful transfer lies in preserving as much of the pretrained prior as possible, eliminating the need to design any complex module or training recipe. Even more notably, DICEPTION is capable of quickly adapting to new tasks using as few as 50 training images and fine-tuning less than 1% of its parameters. We also demonstrate that pixel-level aligned training for perception tasks significantly enhances the model's ability to preserve fine details and mitigates generated artifacts, which is of high significance for downstream applications. We believe DICEPTION provides valuable insights for the design of strong diffusion-based generalist models.

In summary, our main contributions are as follows.

- We introduce **DICEPTION**, to the best of our knowledge, the first unified multi-task perception model with **fully shared parameters** that achieves quantitative performance comparable to specialized models while requiring significantly less data. *E.g.*, we achieve competitive

results with SAM-vit-h with only 0.06% of its data. We are capable of addressing six visual perception tasks within one single model.

- This work offers a comprehensive experimental analysis elucidating the critical designs for effectively re-purposing diffusion models towards perception tasks, including architecture, input injection strategies and sampling timestep selection. Our findings establish that the preservation of the pretrained generative prior is paramount for achieving rapid adaptation and robust multi-task performance. Notably, DiT architectures are shown to be particularly conducive to this objective.

- The proposed unified multi-task paradigm yields compelling advantages. For instance, DICE-PTION rapidly adapts to novel tasks in few-shot settings, demonstrating strong performance with as few as 50 training images and fine-tuning only 1% of parameters. Training on pixel alignment tasks significantly mitigates the artifacts often observed in other generative models for low-level image processing tasks such as image highlighting. Furthermore, the unified prediction space enables interactive segmentation to achieve matting-level accuracy.

## 2 Related Work

### 2.1 Vision Foundation Models

Vision foundation models are models that are trained on large-scale datasets and demonstrate excellent performance within their trained domains. Vision foundation models now exist for a broad range of vision tasks, including monocular depth estimation [125, 126, 123, 7], object detection [11], segmentation [51, 90], multimodal tasks [86, 66], image and video generation [94, 29, 6], and more recently, emerging 3D models [111, 73]. While many works [117, 50, 60, 87, 140, 143] have sought to leverage the prior knowledge embedded in these models to tackle other tasks, such efforts often require complex network designs and intricate training strategies, typically transferring only to a limited number of tasks. Some foundation models [91, 40, 78, 12] emphasize representation learning, aiming to solve diverse downstream tasks by relying on generalized features. However, the results of these methods often fall short when compared with specialized foundation models. In contrast, our approach ensures consistent accuracy across multiple tasks while also enabling swift adaptation to new downstream tasks.

### 2.2 Diffusion Models

Diffusion models [29, 94, 6, 104, 103] have achieved remarkable success in image and video generation in recent years. The idea is to gradually add noise to the data and train a model to reverse this process, denoising step by step to generate the result. Recent diffusion models [29] utilize flow matching [65, 1, 68] and the DiT architecture [80], making them more scalable and efficient. Diffusion models have enabled a wide range of notable applications, including conditional image generation [137, 63, 130, 76, 85], image editing [8, 48, 122], story generation [113, 142], video generation [42, 36, 139, 128, 6, 52, 112], and video editing [13, 67, 14]. These successes underscore the substantial prior knowledge embedded in diffusion models.

Building on this insight, many studies [120, 39, 129, 49, 143] leverage diffusion models for downstream image understanding tasks. However, these approaches typically require separate fine-tuning for each individual task. Recently, we find several concurrent works [118, 55] also use diffusion models for multitask learning. Yet, these methods often involve complex network architectures and training procedures, and their evaluations tend to focus only on a very limited subset of image understanding results. In contrast, our DICEPTION offers a simpler solution. We not only conduct detailed evaluations of our method across a variety of tasks but also demonstrate that the simplicity, paired with the inherent strengths of diffusion models, can be sufficient to deliver strong results without relying on overly complicated setups.

### 2.3 Multi-task Generalist Models

Recently, there has been a surge of interest in exploring visual multitask learning. Some approaches [115, 116] draw inspiration from in-context learning in NLP, adapting it for the visual domain. Others [71, 70, 75, 2] have advocated for sequence modeling methods, utilizing a transformer encoder-decoder architecture. In these approaches, different encoders map various tasks into a shared

representation space, and distinct decoders are employed to transform tokens into the outputs specific to each task. However, these methods face notable limitations: they need to train a separate encoder and decoder for every individual task and they usually rely on substantial amounts of data to attain optimal performance.

The recent success of high-quality Vision Language Models (VLMs) [66] has also encouraged researchers to leverage them for building multitask models. Yet, these VLM-based methods [4, 110, 17, 69, 92, 61] typically focus on multimodal understanding tasks, such as image captioning, rather than general visual perception tasks. Meanwhile, some approaches [101, 139, 79] combine diffusion models with autoregressive models, focusing primarily on instruction-following image generation or editing tasks, rather than addressing image perception tasks. Although certain studies [54, 47, 18, 35] have tried to apply VLMs to more advanced semantic perception tasks, they struggle to establish a unified generalist visual model.

## 2.4 Compared with One Diffusion

The concurrent work, One Diffusion [55], addresses multi-task image generation, whereas our approach focuses on multi-task image understanding. We excel at performing a broader range of image understanding tasks with higher quality. While One Diffusion's strategy of treating different images as different views benefits generation tasks, their failure to distinguish between conditions and images introduces harmful degrees of freedom for perception tasks, as illustrated in the red-highlighted regions of Figure S14. Specifically, when performing perception tasks, One Diffusion tends to generate an image similar to the original input, rather than the desired perceptual results.

Although One Diffusion suggests that more detailed text prompts can lead to better results, we argue that **performance in perception tasks should not overly depend on the quality of text prompts.** In contrast, our method uses only simple task prompts to distinguish between different tasks, rather than allowing the text prompts to dominate the results.

Crucially, while One Diffusion requires a massive amount of data (75 million samples) and computational resources for from-scratch training, we leverage the priors of pretrained models and demonstrate that, with significantly less data (1.8 million samples), we achieve performance on par with state-of-the-art results. In the image understanding tasks shared by both approaches, we consistently produce more stable and higher-quality results than One Diffusion.

# 3 Method

## 3.1 Overview

Our methodology builds upon pre-trained text-to-image diffusion models [29], steering perception tasks using text prompts. As shown in Figure 2, we concatenate the input image tokens, the noisy tokens, task prompt embeddings, and point embeddings for interactive segmentation along the token dimension. Training employs a flow matching loss [29], exclusively computed on the noisy tokens. In inference, each denoising step refines only these noisy tokens, leaving all other conditioning tokens unchanged throughout the iterative denoising process.

## 3.2 Unifying Task Representation into RGB Space

The decision to unify representations of diverse tasks in RGB space was motivated by two key factors: (1) It maximally leverages the priors in text-to-image models, which have been extensively trained within the RGB domain. (2) RGB serves as a foundational representation in computer vision, providing a common visual framework through which a wide variety of tasks can be coherently and intuitively visualized.

We focus on several of the most fundamental tasks in computer vision: monocular depth estimation, normal estimation, human keypoint estimation and segmentation. Segmentation, in particular, encompasses interactive segmentation, entity segmentation, and instance segmentation. Our instance segmentation segments target instances with category name as input. All these tasks can be unified within an RGB space, with the difference being the number of channels. For single-channel representations, such as depth maps and segmentation masks, we align them with RGB by repeating the

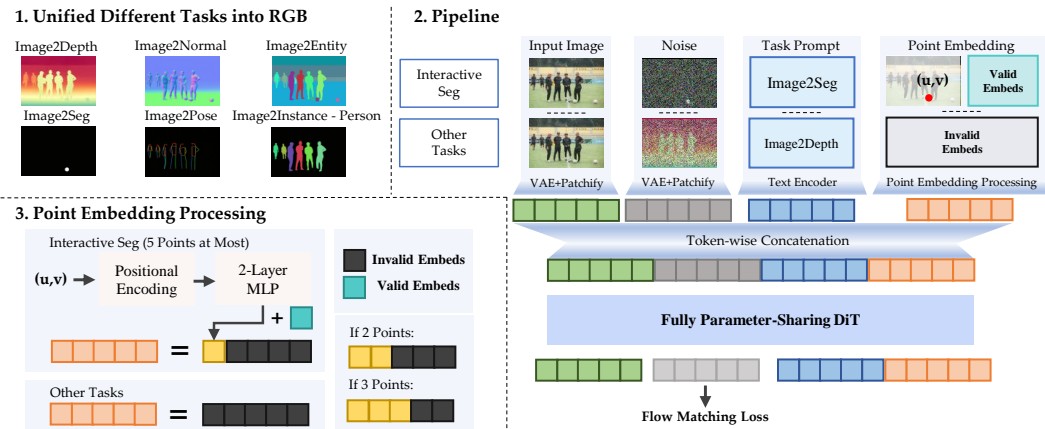

Figure 2: We propose a generalist diffusion model solving multiple perception tasks, **DiCeption**. We select Person as the instance segmentation example for the purpose of consistent visualization, which does not mean our method is limited to only human instances. At each denoising step, the point embedding, input image latent, and task embedding remain fixed, while only the noise latent is updated.

channel three times. For inherently three-channel representations, such as normal maps, we treat them directly as RGB images.

Entity segmentation is to segment every instance in an image but with no category. We assign each mask within an image a random color and merge them into a three-channel RGB mask. Painter [115] found that assigning color randomly makes the model hard to optimize. However, we find this approach has no adverse impact on the training and enables the model to effectively learn to distinguish different instances by painting them with different colors. Each instance's mask can be extracted from the RGB mask using clustering algorithms during post-processing without significant performance degradation. We also apply the random color assignment in instance segmentation. Our method is capable of segmenting instances of the same semantic category. By default, we use KMeans for mask extraction.

Let $\mathbf{x}_r$ denote the pre-unified raw representation for each task, and $\mathbf{x}$ represents the unified RGB-like output representation. We formalize this process as: $\mathbf{x} = \Psi(\mathbf{x}_r)$.

### 3.3 DiCeption: A Unified Framework

**Architecture.** Our model adopts the same architecture as SD3 [29]. We aim to keep the architecture as unchanged as possible, fully leveraging the pre-trained prior knowledge. To do so, we concatenate the input image tokens, noisy tokens, task embeddings, and point embeddings along the token dimension as input to the model. During training, the loss is computed only on the noisy tokens. Similarly, during inference, at each timestep, only the noisy tokens are updated, while the other tokens remain unchanged. We use simple task prompts to direct the model to perform various tasks, such as "image to depth", "image to normal", and "image to segmentation". An additional category name is provided in instance segmentation, such as "image to instance - cat".

**Introduction of Point Embeddings** For point-prompted interactive segmentation, a naive approach is directly painting points on the image. But this strategy is highly sensitive to the size of the points. If the painted points are too large, they can obscure small regions, causing segmentation to fail. Conversely, if the painted points are too small, the model may lose relevant point information after VAE downsampling and patchification. To address this, we introduce a minimal straightforward two-layer MLP $\Phi(\cdot)$ that enables the model to understand the point prompt.

Inspired by SAM [51], we apply sin-cos positional encoding to the point coordinates $p$, then pass them into the MLP $\Phi(\cdot)$ to produce point embeddings that match the dimension of the input hidden states. We use two learnable embeddings to indicate whether the embedding is valid or not: $\xi_p$ for

valid point embeddings and $\xi_{np}$ for invalid point embeddings. The processed point embedding is summed with $\xi_p$. For other tasks, we simply use $\xi_{np}$ as the point embedding. During training, we randomly select 1–5 points to guide the segmentation. When the number of selected points is fewer than 5, we pad the point embeddings to a length of 5 with $\xi_{np}$. When performing tasks that do not require point input, the point embedding is simply a length-5 sequence, where each element is $\xi_{np}$. By denoting the final point embedding as $\xi$, this process is formulated as:

$$\xi = \begin{cases} \text{Concat}(\Phi(\text{PE}(p)) + \xi_p, \xi_{np}) & \text{if interactive segmentation} \\ \xi_{np} & \text{else} \end{cases} \tag{1}$$

**Input Formulation and Loss.** DICEPTION introduces two additional inputs based on SD3: the input image $\mathbf{x}'$ and point embedding $\xi$. For the input image, we first apply VAE to down-sample it by a factor of 8, after which it is $2 \times 2$ patchified into sequences. We denote this pre-processing as $\tau$. Subsequently, the task prompt token $\mathbf{e}$, point embedding $\xi$, noisy token $\mathbf{z}_t$, and input image token $\mathbf{z}'$ are concatenated along the token dimension to form the complete input. We follow the flow matching [65, 1, 68] loss in training SD3 [29], which minimizes the discrepancy between the model's predicted velocity $v$ and the ground-truth velocity $u$. During training, the loss is applied solely to the noisy tokens:

$$\mathbf{z}_0 = \tau(\mathbf{x}), \mathbf{z}' = \tau(\mathbf{x}')$$
$$\text{Loss} = \mathbb{E}_{\mathbf{z}_0, t} \| v_\theta(\mathbf{z}_t, \mathbf{z}', t, \mathbf{e}, \xi) - u(\mathbf{z}_t) \|_2^2. \tag{2}$$

### 3.4 Adapting to New Tasks

Practical applications often require models to adapt quickly to new tasks with limited training data. Traditional foundation models, however, are often domain-specific and require extensive data and architectural modifications for adaptation. Powerful diffusion models also struggle with efficient adaptation to downstream tasks via few-parameter fine-tuning on limited data.

DICEPTION effectively addresses this limitation. We conducted experiments on lung segmentation, tumor segmentation, and image highlighting, which represent tasks with varying degrees of overlap with the model's original domain. We train fewer than 1% of the model's parameters using LoRA [44] without any complex architectural modifications. Notably, despite the limited availability of training samples (50 per task), DICEPTION consistently delivered successful and high-quality performance across all target tasks. These results provide compelling evidence for the potential of DICEPTION as a unified foundation model.

## 4 Experiments

### 4.1 Implementation Details

**Data.** We *randomly* select 500k images from the OpenImages [53] dataset and use DepthPro [7] and StableNormal [129] to generate depth and normal annotations. For interactive segmentation, we *randomly* select 400k images from the SA-1B [51] dataset, as well as 200k images with fine-grained hair masks synthesized from the AM2k [58], AIM500 [59], and P3M-10k [57]. Entity segmentation data is from EntityV2 [84], while instance segmentation data comes from the COCO-Rem [97], and human pose data is sourced from COCO [64]. For few-shot fine-tuning, we select 50 samples from the Chest X-Ray dataset [114], LOL-v2 [127], and Kaggle's Brain Tumor dataset as training samples. More details can be found in Appendix A.

**Training.** Our training lasts for 24 days using 4 NVIDIA H800 GPUs. We employ the AdamW optimizer with a constant learning rate of $2e-5$ and a batch size of 28 per GPU. We found that the training process is highly stable. However, the convergence speed for segmentation tasks was slower compared to depth and normal tasks. Therefore, we increased the proportion of segmentation data in each batch. Specifically, in each batch, depth and normal each account for 15%, interactive segmentation, entity segmentation, and instance segmentation each account for 20%, and pose estimation accounts for 20%. We observe that, by the end of training, despite the loss no longer significantly decreasing, the model's performance on segmentation tasks continues to improve.

During few-shot fine-tuning, we apply a rank-128 LoRA to all attention $Q$, $K$, and $V$ layers in the network, which accounts for less than 1% of the total network parameters. The task prompts for

different tasks are "image-to-segmentation lung," "image-to-segmentation tumor," and "image-to-highlight." LoRA training is conducted on a single NVIDIA H100 GPU, with a constant learning rate of $2e-5$ and a batch size of 8. Please refer to Appendix D for more few-shot fine-tuning visualizations.

**Inference.** We perform 28 steps of denoising during inference which follows the settings of the pre-trained model SD3 [29]. The inference can be run on a GPU of 24GB memory with a batch size of 4. The classifier-free-guidance value is by default set to 2, more analysis in Appendix B.

## 4.2 Comparisons with Existing Methods

Table 1: Quantitative comparison of depth estimation with both specialized models and multi-task models on zero-shot datasets. Our visual generalist model can perform *on par* with SOTA models. We use the same evaluation protocol (†) as Genpercept [120].

| Method | Training Samples | KITTI [33] AbsRel↓ | KITTI [33] $\delta_1$↑ | NYUv2 [77] AbsRel↓ | NYUv2 [77] $\delta_1$↑ | ScanNet [24] AbsRel↓ | ScanNet [24] $\delta_1$↑ | DIODE [108] AbsRel↓ | DIODE [108] $\delta_1$↑ | ETH3D [95] AbsRel↓ | ETH3D [95] $\delta_1$↑ |
|---|---|---|---|---|---|---|---|---|---|---|---|
| MiDaS [89] | 2M | 0.236 | 0.630 | 0.111 | 0.885 | 0.121 | 0.846 | 0.332 | 0.715 | 0.184 | 0.752 |
| Omnidata [28] | 12.2M | 0.149 | 0.835 | 0.074 | 0.945 | 0.075 | 0.936 | 0.339 | 0.742 | 0.166 | 0.778 |
| DPT-large [88] | 1.4M | 0.100 | 0.901 | 0.098 | 0.903 | 0.082 | 0.934 | 0.182 | 0.758 | 0.078 | 0.946 |
| DepthAnything† [125] | 63.5M | 0.080 | 0.946 | 0.043 | 0.980 | 0.043 | 0.981 | 0.261 | 0.759 | 0.058 | **0.984** |
| DepthAnything v2† [126] | 62.6M | 0.080 | 0.943 | 0.043 | 0.979 | 0.042 | 0.979 | 0.321 | 0.758 | 0.066 | 0.983 |
| Depth Pro† [7] | - | 0.055 | 0.974 | 0.042 | 0.977 | 0.041 | 0.978 | 0.217 | 0.764 | 0.043 | 0.974 |
| Metric3D v2† [45] | 16M | **0.052** | **0.979** | **0.039** | 0.979 | **0.023** | **0.989** | 0.147 | **0.892** | **0.040** | 0.983 |
| DiverseDepth [131] | 320K | 0.190 | 0.704 | 0.117 | 0.875 | 0.109 | 0.882 | 0.376 | 0.631 | 0.228 | 0.694 |
| LeReS [132] | 354K | 0.149 | 0.784 | 0.090 | 0.916 | 0.091 | 0.917 | 0.271 | 0.766 | 0.171 | 0.777 |
| HDN [134] | 300K | 0.115 | 0.867 | 0.069 | 0.948 | 0.080 | 0.939 | 0.246 | 0.780 | 0.121 | 0.833 |
| GeoWizard [32] | 280K | 0.097 | 0.921 | 0.052 | 0.966 | 0.061 | 0.953 | 0.297 | 0.792 | 0.064 | 0.961 |
| DepthFM [34] | 63K | 0.083 | 0.934 | 0.065 | 0.956 | - | - | 0.225 | 0.800 | - | - |
| Marigold† [49] | 74K | 0.099 | 0.916 | 0.055 | 0.964 | 0.064 | 0.951 | 0.308 | 0.773 | 0.065 | 0.960 |
| DMP Official† [56] | - | 0.240 | 0.622 | 0.109 | 0.891 | 0.146 | 0.814 | 0.361 | 0.706 | 0.128 | 0.857 |
| GeoWizard† [32] | 280K | 0.129 | 0.851 | 0.059 | 0.959 | 0.066 | 0.953 | 0.328 | 0.753 | 0.077 | 0.940 |
| DepthFM† [34] | 63K | 0.174 | 0.718 | 0.082 | 0.932 | 0.095 | 0.903 | 0.334 | 0.729 | 0.101 | 0.902 |
| Genpercept† [120] | 90K | 0.094 | 0.923 | 0.091 | 0.932 | 0.056 | 0.965 | 0.302 | 0.767 | 0.066 | 0.957 |
| Painter† [115] | 24K | 0.324 | 0.393 | **0.046** | **0.979** | 0.083 | 0.927 | 0.342 | 0.534 | 0.203 | 0.644 |
| Unified-IO† [71] | 48K | 0.188 | 0.699 | 0.059 | 0.970 | **0.063** | **0.965** | 0.369 | 0.708 | 0.103 | 0.906 |
| 4M-XL† [75] | 759M | 0.105 | 0.896 | 0.068 | 0.951 | 0.065 | 0.955 | 0.331 | **0.734** | 0.070 | 0.953 |
| OneDiffusion† [55] | 500K | 0.101 | 0.908 | 0.087 | 0.924 | 0.094 | 0.906 | 0.399 | 0.661 | 0.072 | 0.949 |
| Ours-single† | 500K | 0.064 | 0.952 | 0.066 | 0.953 | 0.077 | 0.942 | 0.283 | 0.717 | 0.052 | 0.971 |
| Ours† | 500K | **0.069** | **0.949** | 0.061 | 0.960 | 0.072 | 0.944 | **0.289** | 0.722 | **0.050** | **0.975** |

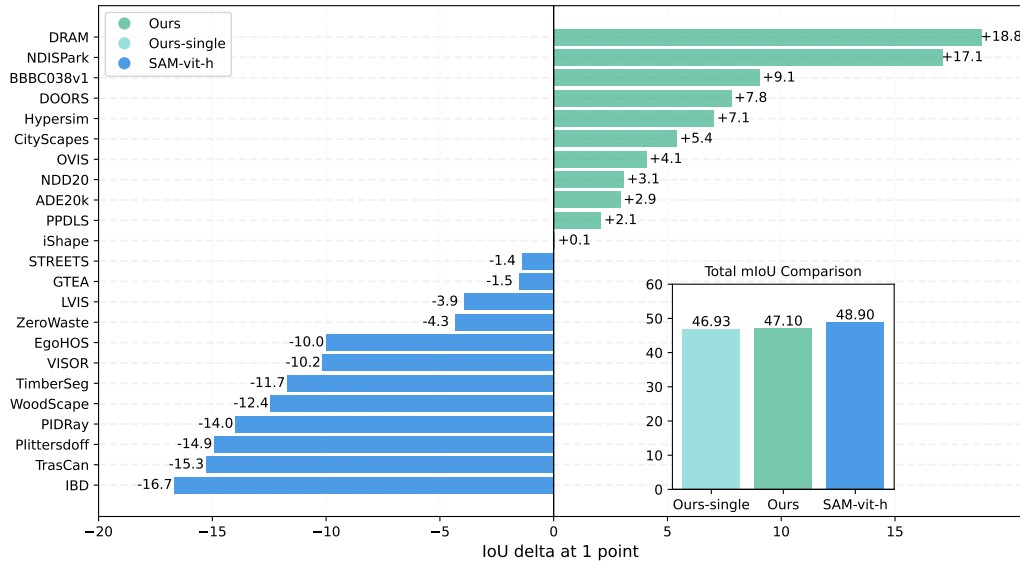

Figure 3: Comparisons of mIoU with SAM-vit-h. **We achieve results on par with SAM using only 0.06% of their data (600K vs. 1B).** The performance of SAM is clearly better only on some datasets that are out-of-distribution for us, such as the Woodscape [133] Fisheye dataset.

Table 2: Quantitative comparison of surface normal estimation with both specialized models and multi-task models. All methods are evaluated with the same method of StableNormal [129].

| Method | Training Samples | NYUv2 [77] | | | | | ScanNet [24] | | | | | DIODE-indoor [108] | | | | |
|---|---|---|---|---|---|---|---|---|---|---|---|---|---|---|---|---|
| | | mean↓ | med↓ | 11.25°↑ | 22.5°↑ | 30°↑ | mean↓ | med↓ | 11.25°↑ | 22.5°↑ | 30°↑ | mean↓ | med↓ | 11.25°↑ | 22.5°↑ | 30°↑ |
| DINSE [3] | 160K | **18.572** | 10.845 | **54.732** | 74.146 | 80.256 | 18.610 | 9.885 | 56.132 | 76.944 | 82.606 | 18.453 | 13.871 | 36.274 | 77.527 | 86.976 |
| Geowizard [32] | 280K | 20.363 | 11.898 | 46.954 | 73.787 | 80.804 | 19.748 | 9.702 | 58.427 | 77.616 | 81.575 | 19.371 | 15.408 | 30.551 | 75.426 | 86.357 |
| GenPercept [120] | 90K | 20.896 | 11.516 | 50.712 | 73.037 | 79.216 | 18.600 | 8.293 | 64.697 | 79.329 | 82.978 | 18.348 | 13.367 | 39.178 | 79.819 | 88.551 |
| Marigold [49] | 90K | 20.864 | 11.134 | 50.457 | 73.003 | 79.332 | 18.463 | 8.442 | 64.727 | 79.559 | 83.199 | 16.671 | 12.084 | 45.776 | 82.076 | 89.879 |
| StableNormal [129] | 250K | 19.707 | **10.527** | 53.042 | **75.889** | **81.723** | 17.248 | 8.057 | **66.655** | **81.134** | **84.632** | 13.701 | 9.460 | 63.447 | 86.309 | 92.107 |
| Unified-IO [70] | 210K | 28.547 | 14.637 | 39.907 | 63.912 | 71.240 | **17.955** | 10.269 | **54.120** | 77.617 | **83.728** | 31.576 | 16.615 | 27.855 | 64.973 | 73.445 |
| 4M-XL [75] | 759M | 37.278 | 13.661 | 44.660 | 60.553 | 65.327 | 30.700 | 11.614 | 48.743 | 68.867 | 73.623 | 18.189 | 12.979 | 36.622 | 81.844 | 87.050 |
| Ours-single | 500K | 18.292 | 10.145 | 52.693 | 76.966 | 83.041 | 18.807 | 10.327 | 52.919 | 75.152 | 82.968 | 16.229 | 11.012 | 50.137 | 83.573 | 88.972 |
| Ours | 500K | **18.338** | **10.106** | **52.850** | **77.079** | **82.903** | 18.842 | **10.266** | 53.610 | 74.895 | 82.864 | **16.297** | **11.117** | **50.548** | **83.325** | **88.774** |

We compare the performance of specialized models, existing multi-task models, and our DI-CEPTION across various tasks. Specifically, we evaluate depth using the same protocol as Gen-percept [120], normal estimation using the same method as StableNormal [129], interactive segmentation using the same approach as SAM [90], and human keypoints using the same method as Painter [115]. We also assess instance segmentation and entity segmentation on the MS COCO dataset. For entity segmentation, we assigned all predicted categories to the same label.

Table 3: Evaluation of human keypoints estimation on MS COCO.

| | HRNet[100] | HRFormer[135] | ViTPose[121] | Painter[115] | Ours |
|---|---|---|---|---|---|
| AP↑ | 76.3 | 77.2 | **78.3** | 72.5 | 57.8 |

As in Tables 1 and 2, our DICEPTION outperforms existing multi-task models and achieves performance on par with state-of-the-art specialized models or demonstrates only an acceptable performance decrease. Although some multi-task methods achieve marginally better performance on certain datasets, such as Painter [115] and Unified-IO [70], they exhibit considerably poorer results on others such as outdoor settings (KITTI) and NYUv2 normal map benchmark. This further underscores the robust generalization capabilities of our approach. We contend that focusing on a model's performance across diverse datasets is more meaningful, as it better reflects the model's generalization ability and real-world applicability.

Table 4: Evaluation of text-based instance segmentation on the MS COCO.

| Method | SparK [102] | OneFormer [46] | Mask2Former [19] | Ours |
|---|---|---|---|---|
| AP↑ | 45.1 | 49.2 | **50.1** | 33.2 |

For interactive segmentation, as shown in Figure 3, **we achieve results on par with SAM-vit-h using only 0.06% of their data.** SAM shows a clear advantage only on certain out-of-distribution datasets that are outside the scope of our model's training, such as WoodScape fisheye dataset. *Notably, while most specialized models require extensive data or complex data pipelines, our method achieves excellent results with significantly less data and no training data cherry-picking.* Evaluation across diverse datasets highlights the strong in-the-wild generalization capability of our model, demonstrating that it does not overfit to the biases inherent in specific datasets.

We observe that, although our model generates high-quality visualizations for human pose and instance segmentation, the corresponding evaluation metrics remain relatively low. This is also observed on the evaluation of small objects in entity segmentation. We found that this is due to the errors introduced by the post-processing rather than our model's performance. In Appendix C, we provide a comprehensive explanation of the post-processing procedure and analyze the underlying causes of metrics degradation.

## 4.3 Ablations and Analysis

**Model designs, classifier-free guidance and pixel-aligned training.** Our crucial analyses covering the elucidation of critical designs for effectively re-purposing diffusion models for perception tasks, as well as significant findings and insights, are detailed in the Appendix due to space limit. Specifically, the analysis of different architectures and input paradigms is presented in Appendix B.1, B.2 and B.3. The effectiveness of modest classifier-free guidance in improving results is discussed in Appendix B.4. The inherent few-step capability of flow-matching on perception tasks is analyzed in Appendix B.5. The benefits of pixel-aligned training are detailed in Appendix B.6 and B.7.

Table 5: Average recall (AR) of entity segmentation on the MS COCO validation set.

| Method | AR-small↑ | AR-medium↑ | AR-large↑ |
|---|---|---|---|
| EntityV2 [84] | **0.313** | **0.551** | **0.683** |
| Ours-single | 0.123 | 0.424 | 0.648 |
| Ours | 0.121 | 0.439 | 0.637 |

**Comparisons with Our Single-task Models.** For the training of single-task models, we ensure that the network architecture remains the same and the total amount of training data seen for each specific task is the same as that for the multi-task model. For example, if the multi-task model is trained for 100 iterations with 4 depth data samples per batch, the single-task model will also be trained for 100 iterations with 4 data samples per batch. In our current data setting (approximately 1.8 million samples), we have not observed a significant gap between the multi-task and single-task models, nor have we seen a trend of mutual promotion between different tasks, as shown by "Ours-single" in Tables 1, 2, 5 and Figure 3. We believe that it is more appropriate to explore with larger datasets in order to draw more solid conclusions. We leave this as future work.

**Multi-point Prompted Segmentation.** Ambiguity is a significant issue in interactive segmentation. For example, if a point is placed on a person's clothing, the model may segment the clothing, but the desired result is the person. Therefore, more points are needed to resolve this ambiguity. As illustrated in Table 6, additional points help the model better segment the desired results.

**One-step Training and One-step Inference.** Genpercept [120] demonstrates that diffusion model trained with one-step denoising significantly enhances both the speed and accuracy of perceptual tasks. However, our experimental results reveal a notable increase of failure cases when applying one-step diffusion in a multi-task setting, as illustrated in Figure 4. We believe that this is due to the potential overlap of denoising trajectories for different

Table 6: Comparisons between 1-point and 5-point as input. 5 points are selected randomly.

| Method | 1-point | 5-point |
|---|---|---|
| mIoU↑ | 47.1 | 57.2 |

tasks. These overlapping trajectories can interfere with each other, resulting in failure cases with one-step inference. In contrast, in a single-task setting, since the denoising trajectories pertain to a single task, one-step is more effective and stable. However, we observe that our model, trained with multi-step denoising, can be applied directly to few-step inference with minimal degradation in performance. We provide results and more detailed analysis in Appendix B.5.

## 5 Conclusion

We have introduced DICEPTION, a multi-task visual generalist model based on the diffusion model. Our approach unifies different tasks in the RGB space, leveraging the prior knowledge of pre-trained image generation model to achieve results that are on par with specialized foundation models. We achieve good performance without carefully cherry-picking extremely high-quality data or by using an exceptionally large amount of data. In few-shot fine-tuning, we are able to achieve high-quality results with minimal data and minimal trainable parameters.

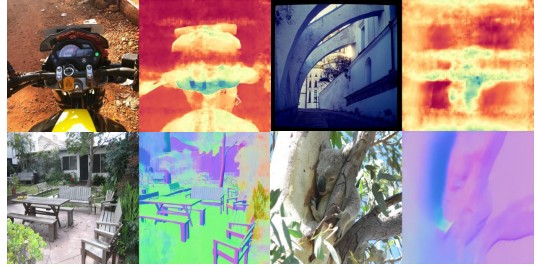

Figure 4: The model trained with 1-step denoising tends to produce more failure cases in multi-task scenarios.

Furthermore, we provide in-depth experimental analyses of strategies for transferring diffusion models to perception tasks. We also discuss the contributions of classifier-free guidance in enhancing model performance, demonstrate that there is no performance gap between our single-task and multi-task model, and highlight the improved detail preservation achieved through pixel-aligned perception training. We believe that DICEPTION sheds light on how to effectively use priors of diffusion models to build a strong visual generalist model.

**Acknowledgement.** This work was in part supported by The Pioneer R&D Program of Zhejiang (Grant No. 2025C01011).

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

# Appendix

## A Dataset

We summarize the datasets used in our work in Table S1. The depth and normal data samples are obtained by randomly selecting 500K images from OpenImages [53] and labeling them using Depth Pro [7] and StableNormal [129], respectively. The 400K point segmentation data samples are obtained by randomly selecting images from the SA-1B dataset [51]. For the synthesis of point segmentation data, we extract the foreground from P3M-10K [57], AIM500 [59] and AM2K [58], randomly applying transformations such as rotation, resizing, and flipping. These transformed foregrounds are then pasted onto different background images, resulting in 200K synthetic images with fine-grained hair segmentation.

Table S1: Dataset detail.

| Training | | |
|---|---|---|
| Task | Data Samples | Dataset |
| Depth | 500K | OpenImages [53] + Depth Pro [7] |
| Normal | 500K | OpenImages [53] + StableNormal [129] |
| Point Segmentation | 400K | SA-1B [51] |
| Point Segmentation | 200K | P3M-10K [57], AIM500 [59] and AM2K [58] |
| Human Pose | 42K | MS COCO 2017 [64] |
| Semantic Segmentation | 120K | COCO-Rem [97] |
| Entity Segmentation | 32K | EntityV2 [84] |
| Validation | | |
| Task | Dataset | |
| Depth | NYUv2 [77], KITTI [33], ScanNet [24], DIODE [108], ETH3D [95] | |
| Normal | NYUv2 [77], ScanNet [24], DIODE [108] | |
| Point Segmentation | PPDLS [74], DOORS [82], TimberSeg [31], NDD20 [107] STREETS [98], iShape [124], ADE20K [141], OVIS [83] Plittersdorf [38], EgoHOS [136], IBD [15], WoodScape [133] TrashCan [43], GTEA [30, 62], NDISPark [21, 20], VISOR [25, 26] LVIS [37], Hypersim [93], Cityscapes [23], DRAM [22] BBBC038v1 [10], ZeroWaste [5], PIDRay [109] | |
| Entity Segmentation | MS COCO 2017 [64] | |
| Semantic Segmentation | MS COCO 2017 [64] | |
| Human Keypoints | MS COCO 2017 [64] | |

For the validation set, we evaluate depth using the same evaluation protocol as Genpercept [120], conducting tests on the NYUv2 [77], KITTI [33], ScanNet [24], DIODE [108], ETH3D [95]. Similarly, for normal estimation, we follow the evaluation protocol of StableNormal [129] and perform evaluations on the NYUv2 [77], ScanNet [24], DIODE [108]. For interactive segmentation, we conduct extensive comparisons across 23 datasets. The remaining tasks, including Entity Segmentation, Instance Segmentation, and Human Keypoints, are evaluated on the MS COCO 2017 dataset [64]. We believe the comprehensive experiments on **over 30 datasets in total** provide solid evidence of the remarkable performance of our method.

## B Additional Analysis

### B.1 Token-wise Concat and Channel-wise Concat

We investigated two distinct methodologies for integrating an auxiliary input image into a Diffusion Transformer (DiT) architecture. The first approach involved concatenating the input image tokens with the noisy image tokens along the token dimension, subsequently feeding this combined sequence directly into the DiT model. The second strategy employed channel-wise concatenation of these inputs, followed by a shallow, two-layer Multi-Layer Perceptron (MLP) to align the channel dimensions with the DiT's input.

Constrained by available computational resources, our analysis is conducted within 2 tasks: depth and surface normal estimation. The datasets utilized for depth and surface normal prediction in this ablation are identical to those specified in Table S1. All training hyperparameters remain consistent across both approaches, with the sole architectural divergence being the aforementioned two-layer MLP utilized for feature alignment in the channel-wise concatenation method.

Our findings indicate that the token-wise concatenation strategy is markedly more computationally efficient than its channel-wise counterpart. Specifically, the token-wise approach demonstrates substantially faster convergence speed, as illustrated by the training loss trajectories presented in Figure S1. Furthermore, as demonstrated in Figure S2, channel-wise concatenation is more prone to yielding suboptimal results. We believe that this enhanced efficiency and effectiveness stem from the token-wise concatenation method's circumvention of additional network parameters. By avoiding the introduction of new trainable components, this strategy appears to more effectively leverage the inherent priors learned by the pre-trained diffusion model. Furthermore, for token-wise concatenation, we independently applied Rotary Position Embeddings (RoPE) [99] to both the input image tokens and the noisy tokens. This strategy ensures that corresponding tokens from these two sources share identical positional embeddings, facilitating the model's rapid learning of their interrelations.

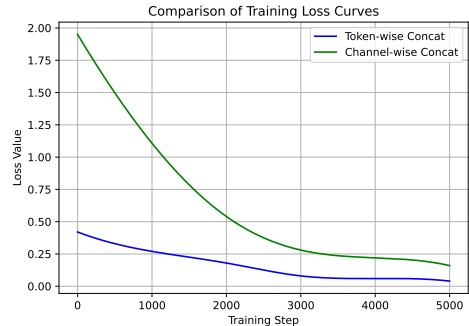

Figure S1: Loss curve of token-wise concatenation and channel-wise concatenation.

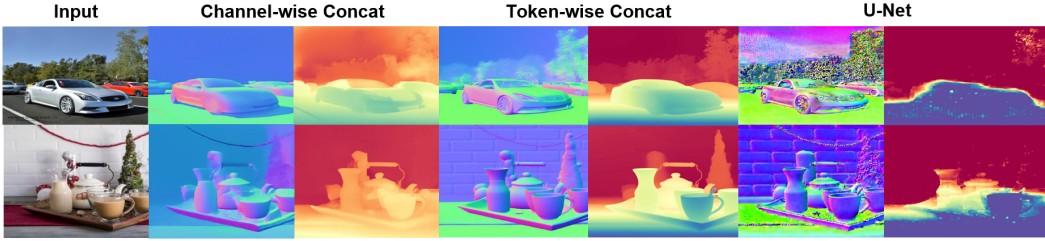

Figure S2: Depth and normal estimation multi-task visualizations comparing channel-wise concatenation, token-wise concatenation, and U-Net are shown. While channel-wise concatenation often leads to suboptimal performance and U-Net struggles with multi-task learning, DICEPTION effectively generates high-quality outputs for multiple tasks.

## B.2  Architecture of the Diffusion Model

Before the advent of DiT [80], the UNet architecture was predominantly used in diffusion models. We also conduct multi-task experiments based on a UNet pre-trained model SDXL [81]. Specifically, we follow Marigold [49] by expanding the first convolution layer's input channels from 4 to 8 to accommodate image inputs, and similarly use task prompts to guide the model in solving different tasks. However, as shown in Figures S2 and S3 , we find that this approach failed, even for a minimal multi-task scenario involving only depth and normal estimation.

Beyond the established UNet architecture, our research also encompasses an exploration of alternative DiT frameworks, notably PixArt-alpha [16], to ascertain the generalizability and efficacy of our proposed methodology when applied to different DiT models. We train DICEPTION-PixArt based on the PixArt-alpha-600M model using the same data for training DICEPTION and conduct a quantitative evaluation on depth and surface normal prediction, as illustrated in Tables S2, S3 and S4.

It is pertinent to note that, with a parameter count of approximately 600M, the DICEPTION-PixArt variant, while not achieving the same performance benchmarks as our counterpart model trained on

Table S2: Quantitative comparison of depth estimation between ours and Ours-PixArt.

| Method | Training Samples | KITTI [33] AbsRel↓ | δ₁↑ | NYUv2 [77] AbsRel↓ | δ₁↑ | ScanNet [24] AbsRel↓ | δ₁↑ | DIODE [108] AbsRel↓ | δ₁↑ | ETH3D [95] AbsRel↓ | δ₁↑ |
|---|---|---|---|---|---|---|---|---|---|---|---|
| Ours | 500K | **0.069** | **0.949** | **0.061** | **0.960** | **0.072** | **0.944** | **0.289** | **0.722** | **0.050** | **0.975** |
| Ours-PixArt | 500K | 0.093 | 0.905 | 0.096 | 0.905 | 0.101 | 0.901 | 0.282 | 0.709 | 0.071 | 0.944 |

the more extensive SD3 architecture, still exhibits a strong capacity for multi-task problem-solving. This multi-tasking proficiency is substantially superior to that of traditional UNet-based models. This result substantiates the versatility of our method and its compatibility with modern transformer-based diffusion models, even with smaller models.

Table S3: Quantitative comparison of surface normal estimation between ours and ours-PixArt.

| Method | Training Samples | NYUv2 [77] mean↓ | med↓ | 11.25°↑ | 22.5°↑ | 30°↑ | ScanNet [24] mean↓ | med↓ | 11.25°↑ | 22.5°↑ | 30°↑ | DIODE-indoor [108] mean↓ | med↓ | 11.25°↑ | 22.5°↑ | 30°↑ |
|---|---|---|---|---|---|---|---|---|---|---|---|---|---|---|---|---|
| Ours | 500K | **18.338** | **10.106** | **52.850** | **77.079** | **82.903** | **18.842** | **10.266** | **53.610** | **74.895** | **82.864** | **16.297** | **11.117** | **50.548** | **83.325** | **88.774** |
| Ours-PixArt | 500K | 20.487 | 12.393 | 48.663 | 72.342 | 80.244 | 21.663 | 14.419 | 37.043 | 70.781 | 79.786 | 17.986 | 11.190 | 50.276 | 79.316 | 85.248 |

Regarding the challenges encountered with UNet-based architectures in multi-task learning paradigms, we posit that their limitations are fundamentally due to two key factors. Firstly, the approach of expanding the input convolution layer introduces additional parameters, thereby potentially disrupting the original model's inherent prior knowledge. Secondly, the downsampling operations within the U-Net architecture result in a significant loss of information.

Table S4: Comparisons of 1-point interactive segmentation between ours and ours-PixArt.

| Method | Ours-PixArt | Ours | SAM-vit-h |
|---|---|---|---|
| mIoU↑ | 40.93 | 47.10 | **48.90** |

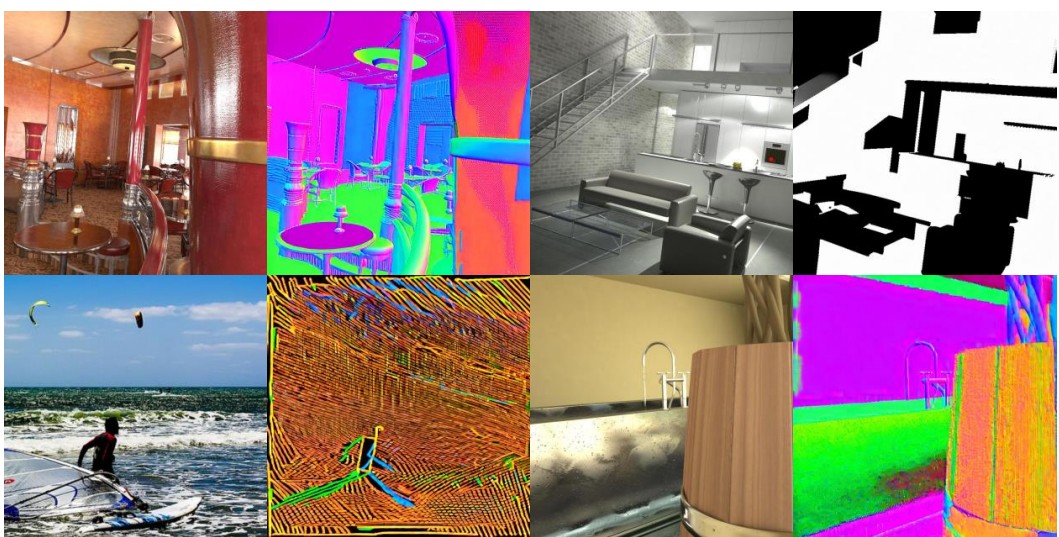

Figure S3: The UNet-based model fails to perform multi-task.

## B.3 ControlNet

ControlNet [137] has emerged as a popular approach for integrating novel image conditioning into diffusion models. However, our experiment shows that while ControlNet can learn the general output patterns associated with target tasks, its precision remains notably low, exhibiting limited performance even on single perception task. We train a ControlNet on top of a pre-trained SD3 model for human keypoint estimation. Following the setup of traditional setting [137], we introduce ControlNet into the first half of the SD3's transformer blocks. As depicted in Figure S4, although the model successfully captures the overall visual style of human keypoint predictions, the accuracy of its estimations is significantly deficient.

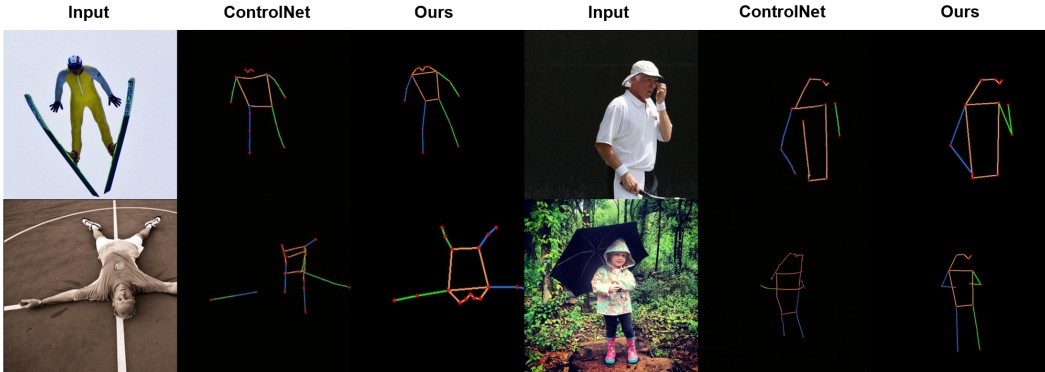

Figure S4: While ControlNet demonstrates the ability to learn the output modalities of perception tasks, its accuracy remains significantly low. Conversely, our proposed approach yields substantially improved accuracy.

## B.4 Classifier-free Guidance

Classifier-free guidance (CFG) [41] is a technique used in conditional diffusion models to improve the quality of generated samples without additional training. It has become a cornerstone in existing text-to-image models. During inference, it extrapolates from the model's conditional and unconditional outputs to enhance the influence of the conditioning signal. Specifically, during denoising, the noise at each timestep is a fusion of conditional and unconditional noise:

$$n_t = n_{t,uncond} + \text{CFG} \cdot (n_{t,cond} - n_{t,uncond}). \tag{S1}$$

Typically, conditional noise $n_{t,cond}$ is the output predicted by the model when conditioned on the prompt embedding, while unconditional noise $n_{t,uncond}$ is the output predicted by the model when conditioned on the negative prompt embedding.

We evaluate the impact of varying CFG values on our multi-task performance. Specifically, our conditional noise $n_{t,cond}$ is the prediction of the model conditioned on the task prompt corresponding to each specific task, while the unconditional noise $n_{t,uncond}$ is the model's prediction when conditioned on an empty string as the prompt. Our ablation study reveals that a modest application of CFG enhances the quality of depth and normal estimation, yielding perceptibly sharper results. However, this strategy basically has no influence on other tasks such as human keypoints estimation and segmentation, as shown in Figure S5..

Table S5: Interactive Segmentation mIoU of DICEPTION across different CFG. CFG has little influence on segmentation.

|  | CFG= 1 | CFG= 2 | CFG= 3 | CFG= 4 | CFG= 5 |
|---|---|---|---|---|---|
| mIoU of 23 Validation Datasets | 47.10 | 47.12 | 47.08 | 46.91 | 46.57 |

We hypothesize that this is because tasks such as depth and normal estimation inherently demand high precision in the output pixel values to accurately represent continuous geometric surfaces, while other tasks such as human keypoints estimation and segmentation are less sensitive to subtle variations in pixel-level intensities. Additionally, it is also observed that a high CFG scale significantly degrades performance on depth and normal prediction, especially normal prediction. This degradation typically manifests as oversaturated results or the emergence of coarse, granular artifacts, as shown in Figure S5. To further validate our hypothesis, we evaluate the performance of our model across varying CFG values, as presented in the Table S6, S7 and S5. The results confirm that a mild CFG scale enhances prediction quality of depth and normal, whereas larger values adversely affect performance.

Table S6: Quantitative comparison of depth estimation with different CFG value.

| Method | Training Samples | KITTI [33] AbsRel↓ | δ₁↑ | NYUv2 [77] AbsRel↓ | δ₁↑ | ScanNet [24] AbsRel↓ | δ₁↑ | DIODE [108] AbsRel↓ | δ₁↑ | ETH3D [95] AbsRel↓ | δ₁↑ |
|---|---|---|---|---|---|---|---|---|---|---|---|
| Ours-CFG=1 | 500K | 0.075 | 0.945 | 0.072 | 0.939 | 0.075 | 0.938 | **0.243** | **0.741** | 0.053 | 0.967 |
| Ours-CFG=2 | 500K | **0.069** | **0.949** | **0.061** | **0.960** | **0.072** | **0.944** | 0.289 | 0.722 | **0.050** | **0.975** |
| Ours-CFG=3 | 500K | 0.092 | 0.910 | 0.076 | 0.938 | 0.093 | 0.910 | 0.343 | 0.679 | 0.059 | 0.966 |
| Ours-CFG=4 | 500K | 0.105 | 0.876 | 0.087 | 0.915 | 0.104 | 0.884 | 0.362 | 0.654 | 0.066 | 0.956 |
| Ours-CFG=5 | 500K | 0.124 | 0.831 | 0.097 | 0.893 | 0.115 | 0.863 | 0.383 | 0.609 | 0.072 | 0.947 |

Table S7: Quantitative comparison of surface normal estimation with different CFG value.

| Method | Training Samples | NYUv2 [77] mean↓ | med↓ | 11.25°↑ | 22.5°↑ | 30°↑ | ScanNet [24] mean↓ | med↓ | 11.25°↑ | 22.5°↑ | 30°↑ | DIODE-indoor [108] mean↓ | med↓ | 11.25°↑ | 22.5°↑ | 30°↑ |
|---|---|---|---|---|---|---|---|---|---|---|---|---|---|---|---|---|
| Ours-CFG=1 | 500K | **18.302** | 10.538 | 52.533 | 75.977 | 82.573 | 19.348 | 12.129 | 46.410 | 74.805 | 82.176 | 17.946 | **8.686** | **62.641** | 81.152 | 85.398 |
| Ours-CFG=2 | 500K | 18.338 | **10.106** | **52.850** | **77.079** | **82.903** | **18.842** | **10.266** | **53.610** | **74.895** | **82.864** | **16.297** | 11.117 | 50.548 | **83.325** | **88.774** |
| Ours-CFG=3 | 500K | 19.817 | 10.989 | 51.312 | 72.509 | 79.497 | 22.287 | 11.849 | 49.110 | 70.075 | 77.376 | 18.546 | 12.475 | 46.627 | 76.532 | 85.398 |
| Ours-CFG=4 | 500K | 21.433 | 12.012 | 47.543 | 69.175 | 77.003 | 24.117 | 13.029 | 41.334 | 65.865 | 73.278 | 22.886 | 14.784 | 41.271 | 65.661 | 74.098 |
| Ours-CFG=5 | 500K | 23.352 | 13.259 | 43.016 | 65.727 | 73.443 | 26.972 | 14.364 | 35.419 | 57.822 | 68.776 | 27.046 | 19.286 | 33.349 | 56.885 | 66.728 |

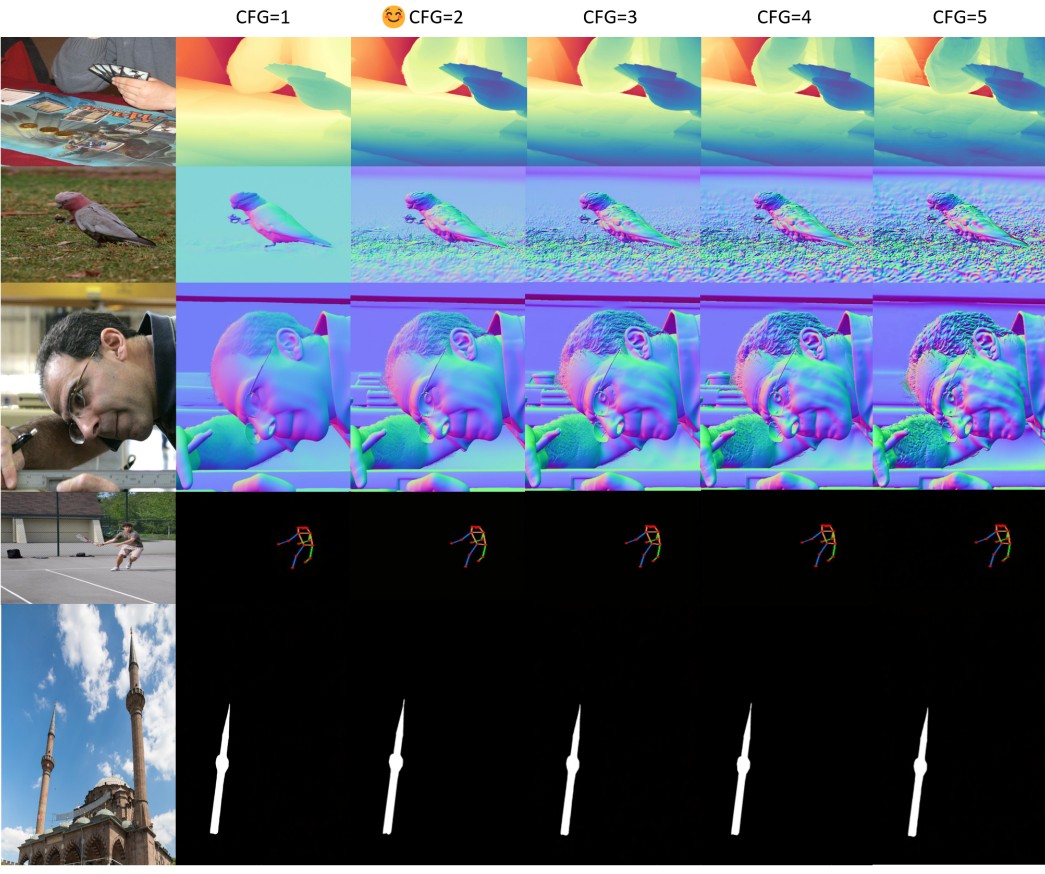

Figure S5: Results on different guidance scale. Depth and normal predictions are highly sensitive to the CFG value, whereas other tasks are barely affected. Based on both the visualization results and the evaluation metrics in Tables S6, S7 and S5, we set the CFG value to 2 by default.

## B.5  Flow-matching Inherently Support Few-step Inference In Perception

We conduct experiments and observe that our model inherently supports few-step inference for perception tasks without any additional techniques, including classifier free guidance, and shows very little performance degradation. The effectiveness of few-step acceleration varies across different tasks. For tasks such as depth and surface normal estimation, the number of inference steps can be reduced to as few as one with acceptable slight performance degradation. For more complex tasks such as interactive segmentation, the model is still able to achieve comparable results using significantly fewer steps while maintaining competitive performance, as demonstrated in Tables S8, S9, and S10. *To the best of our knowledge, this is the first time such a capability is demonstrated in diffusion model for multi-task perception. It strongly supports the advantage of flow-matching-based diffusion models in solving perception tasks.*

Table S8: Quantitative comparison of our few-step depth estimation results.

| Method | Training Samples | KITTI [33] AbsRel$\downarrow$ | $\delta_1\uparrow$ | NYUv2 [77] AbsRel$\downarrow$ | $\delta_1\uparrow$ | ScanNet [24] AbsRel$\downarrow$ | $\delta_1\uparrow$ | DIODE [108] AbsRel$\downarrow$ | $\delta_1\uparrow$ | ETH3D [95] AbsRel$\downarrow$ | $\delta_1\uparrow$ |
|---|---|---|---|---|---|---|---|---|---|---|---|
| 28-step | 500K | 0.069 | 0.949 | 0.061 | 0.960 | 0.072 | 0.944 | 0.289 | 0.722 | 0.050 | 0.975 |
| 14-step | 500K | 0.077 | 0.942 | 0.063 | 0.958 | 0.074 | 0.943 | 0.272 | 0.718 | 0.048 | 0.978 |
| 7-step | 500K | 0.081 | 0.939 | 0.065 | 0.953 | 0.078 | 0.943 | 0.286 | 0.714 | 0.052 | 0.971 |
| 3-step | 500K | 0.083 | 0.938 | 0.069 | 0.953 | 0.077 | 0.940 | 0.294 | 0.707 | 0.063 | 0.967 |
| 1-step | 500K | 0.086 | 0.936 | 0.072 | 0.945 | 0.076 | 0.937 | 0.305 | 0.702 | 0.065 | 0.967 |

Table S9:  Quantitative comparison of our few-step normal map results.

| Method | Training Samples | NYUv2 [77] mean$\downarrow$ | med$\downarrow$ | 11.25°$\uparrow$ | 22.5°$\uparrow$ | 30°$\uparrow$ | ScanNet [24] mean$\downarrow$ | med$\downarrow$ | 11.25°$\uparrow$ | 22.5°$\uparrow$ | 30°$\uparrow$ | DIODE-indoor [108] mean$\downarrow$ | med$\downarrow$ | 11.25°$\uparrow$ | 22.5°$\uparrow$ | 30°$\uparrow$ |
|---|---|---|---|---|---|---|---|---|---|---|---|---|---|---|---|---|
| 28-step | 500K | 18.338 | 10.106 | 52.850 | 77.079 | 82.903 | 18.842 | 10.266 | 53.610 | 74.895 | 82.864 | 16.297 | 11.117 | 50.548 | 83.325 | 88.774 |
| 14-step | 500K | 18.631 | 10.463 | 52.837 | 75.288 | 81.682 | 18.337 | 10.579 | 53.223 | 75.533 | 82.631 | 16.131 | 11.463 | 50.849 | 83.391 | 88.829 |
| 7-step | 500K | 18.335 | 10.492 | 52.771 | 75.443 | 81.936 | 19.008 | 10.363 | 52.628 | 74.886 | 82.055 | 16.835 | 11.330 | 50.039 | 82.443 | 88.218 |
| 3-step | 500K | 18.067 | 10.417 | 53.046 | 76.500 | 81.673 | 19.337 | 10.329 | 52.223 | 75.731 | 82.081 | 17.205 | 12.047 | 50.046 | 83.010 | 87.531 |
| 1-step | 500K | 18.094 | 10.382 | 51.839 | 76.575 | 81.371 | 19.386 | 10.395 | 52.139 | 75.492 | 81.879 | 17.004 | 11.849 | 49.808 | 82.972 | 87.582 |

Table S10: Interactive Segmentation mIoU of DICEPTION across different inference steps.

|  | 28-step | 14-step | 7-step | 3-step | 1-step |
|---|---|---|---|---|---|
| mIoU of 23 Validation Datasets | 47.10 | 47.01 | 46.89 | 45.18 | 42.53 |

We believe that this is because flow matching explicitly imposes linear constraints at each intermediate denoising step—specifically, each noisy latent is constructed as a linear interpolation between the pure noise and the target signal. This process effectively straightens the denoising trajectory, allowing the model to follow an approximately linear path even when using only a few inference steps. In contrast, if the model is trained solely with one-step denoising, the intermediate steps are not constrained and lacks this linear constraint across the trajectory, thus producing poor results as we show in Section 4.3. In contrast, traditional ODE-based diffusion models do not impose such linear trajectory constraints, and therefore cannot support inference with few denoising steps (such as 4 steps) after being trained with multi-step denoising (such as 50 steps). Our additional experiment proves this. We further experiment with PixArt-alpha [16], which uses a DiT-style architecture but adopts a standard ODE-based scheduler. Its results significantly deteriorate when the number of inference steps is reduced, as shown in Table S11, further supporting our analysis.

In image generation tasks, simply reducing inference steps in a flow-matching-based text-to-image model also leads to noticeable quality degradation. This is due to the high complexity and variability introduced by diverse text prompts. In contrast, our perception tasks eliminate the influence of textual prompts, which we believe explains why prior works like One Diffusion [55] require 50 100 inference steps for denoising while ours works well with just a few steps. For comparisons on inference efficiency, we select One Diffusion as baseline and conduct a comparative study on our shared task, depth estimation, under varying numbers of inference steps, as demonstrated in Table S12. Unlike One Diffusion, which suffers from significant performance degradation during few-step inference and fails to produce reasonable results in the 1-step setting, our method is capable of generating high-quality outputs even with just a single inference step. The results demonstrate that our method significantly outperforms One Diffusion in both efficiency and output quality.

Table S11: Quantitative comparison of few-step depth estimation results using Pixart-alpha.

| Method | Training Samples | KITTI [33] AbsRel↓ | KITTI [33] $\delta_1$↑ | NYUv2 [77] AbsRel↓ | NYUv2 [77] $\delta_1$↑ | ScanNet [24] AbsRel↓ | ScanNet [24] $\delta_1$↑ | DIODE [108] AbsRel↓ | DIODE [108] $\delta_1$↑ | ETH3D [95] AbsRel↓ | ETH3D [95] $\delta_1$↑ |
|---|---|---|---|---|---|---|---|---|---|---|---|
| 20-step | 500K | 0.093 | 0.905 | 0.096 | 0.905 | 0.101 | 0.901 | 0.282 | 0.709 | 0.071 | 0.944 |
| 10-step | 500K | 0.146 | 0.872 | 0.153 | 0.861 | 0.159 | 0.844 | 0.347 | 0.658 | 0.119 | 0.895 |

Table S12: Quantitative comparison of One Diffusion and DICEPTION in few-step depth estimation. We compared three experimental settings based on the number of steps: the default configuration, a quarter of the default steps, and one single step.

| Method | KITTI [33] AbsRel↓ | KITTI [33] $\delta_1$↑ | NYUv2 [77] AbsRel↓ | NYUv2 [77] $\delta_1$↑ | ScanNet [24] AbsRel↓ | ScanNet [24] $\delta_1$↑ | DIODE [108] AbsRel↓ | DIODE [108] $\delta_1$↑ | ETH3D [95] AbsRel↓ | ETH3D [95] $\delta_1$↑ |
|---|---|---|---|---|---|---|---|---|---|---|
| Ours-28-step | 0.069 | 0.949 | 0.061 | 0.960 | 0.072 | 0.944 | 0.289 | 0.722 | 0.050 | 0.975 |
| Ours-7-step | 0.081 | 0.939 | 0.065 | 0.953 | 0.078 | 0.943 | 0.286 | 0.714 | 0.052 | 0.971 |
| Ours-1-step | 0.086 | 0.936 | 0.072 | 0.945 | 0.076 | 0.937 | 0.305 | 0.702 | 0.065 | 0.967 |
| OD-50-step | 0.101 | 0.908 | 0.087 | 0.924 | 0.094 | 0.906 | 0.399 | 0.661 | 0.072 | 0.949 |
| OD-12-step | 0.142 | 0.867 | 0.114 | 0.871 | 0.128 | 0.853 | 0.411 | 0.659 | 0.092 | 0.910 |
| OD-1-step | FAIL | FAIL | FAIL | FAIL | FAIL | FAIL | FAIL | FAIL | FAIL | FAIL |

## B.6 Few-shot Finetuning Comparisons on SD3 and Ours

We conduct a comparative evaluation of few-shot tuning performance between SD3 and our DICEPTION. All training data and settings are kept identical for both approaches to ensure a fair comparison. Our findings reveal that DICEPTION not only adapts to new tasks more rapidly but also achieves better performance post-convergence when compared to SD3. Specifically, Figure S6 (a) illustrates that after convergence, our method yields higher-quality results than SD3 on image highlighting. Furthermore, as depicted in Figure S6 (b), DICEPTION demonstrates faster convergence speed. These results collectively underscore the substantial potential of our model for efficient and effective adaptation to novel tasks.

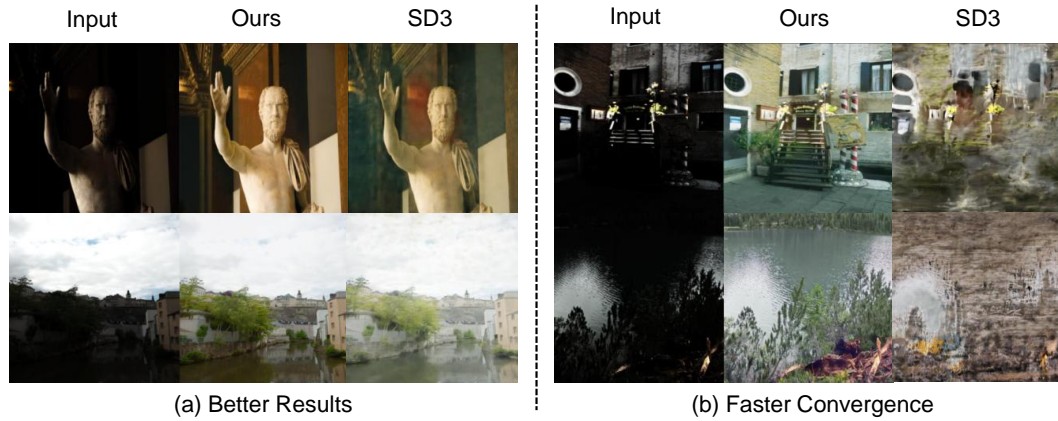

(a) Better Results          (b) Faster Convergence

Figure S6: Image highlighting few-shot finetuning comparisons on SD3 and Ours. (a) Our DICEPTION achieves better performance. Pixel-level aligned training mitigates generated artifacts. (b) Results of Ours and SD3 in the same training iteration. Our DICEPTION is able to adapt to new tasks faster than SD3.

## B.7 Pixel-Level Alignment Training Enhances Detail Preservation

We find that training on pixel-level aligned perception tasks endows the model with a strong ability to preserve fine-grained details. We argue that this capability holds significant practical value. For instance, while existing state-of-the-art method IC-Light [138] for image relighting can generate visually impressive results, it often suffers from noticeable detail loss such as inconsistency of the individuals' appearance. In contrast, our approach demonstrates superior fidelity in preserving

fine-grained details, including nuances that may not be readily perceptible to the human eye. This is demonstrated in our qualitative comparisons in Figure S7.

*It is important to emphasize that our goal is not to compare the lighting quality between methods, but rather to highlight our model's ability to significantly reduce generative artifacts and retain structural details.* We attribute this strength to the model's exposure to pixel-level aligned tasks during training. Additional comparisons with SD3 [29] in Figure S6 further support this observation. We consider this finding highly promising and believe it holds substantial implications for detail-preserving generative modeling and downstream applications.

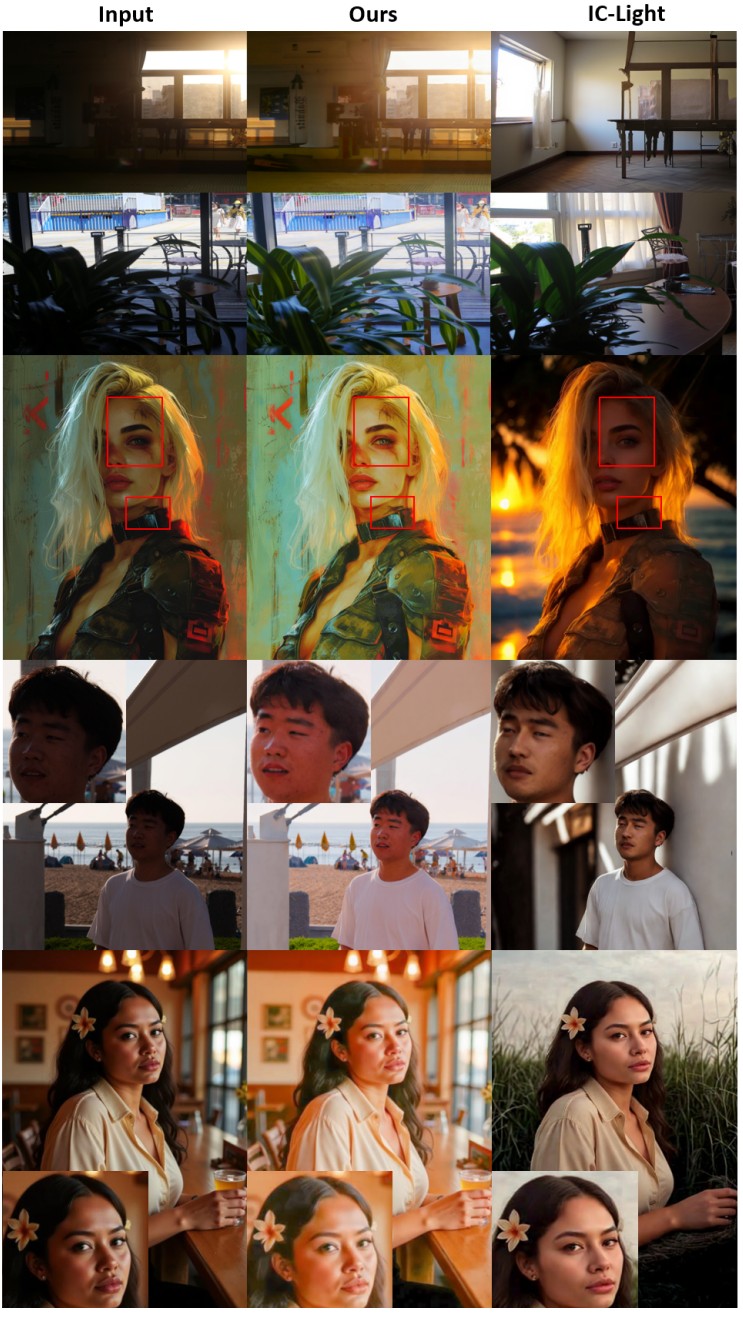

Figure S7: **Comparisons of detail preservation, rather than lighting quality.** Pixel-level aligned training leads to improved preservation of fine-grained details. Better viewed with zoom-in. Input images are generated and from the public BAID dataset [72].

## C    Post-processing

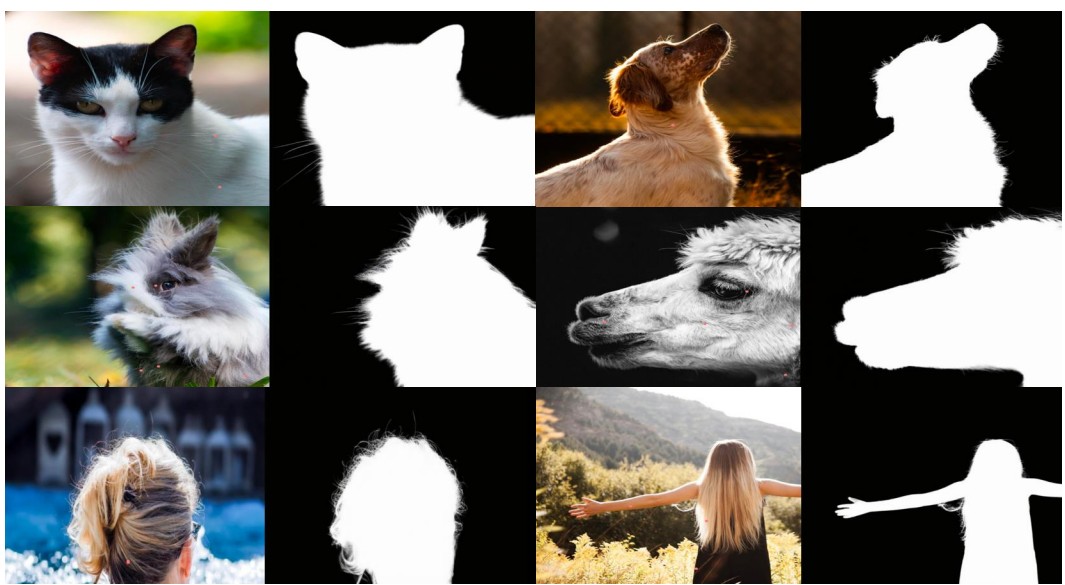

Figure S8: Segmentation results on furry objects. Our interactive segmentation achieves matting-level accuracy.

---

**Algorithm 1** Keypoints Post-processing

---

**Input:** human pose RGB $\mathbf{x}$, GT keypoints $\mathbb{K}_{gt}$, RGB tolerance $\sigma$, distance threshold $\xi$
**Output:** extracted keypoints $\mathbb{K}_{pred}$
1: $\mathbf{x}' = \text{ExtractRedRegions}(\mathbf{x}, (255, 0, 0), \sigma)$
2: $\mathbf{x}_c = \text{GetConnectedComponents}(\mathbf{x}')$
3: $\mathbb{C} = \text{GetCircular}(\mathbf{x}_c)$
4: $\mathbb{K}_{pred} = \varnothing$
5: **for** $\mathbf{c} \in \mathbb{C}$ **do**
6: $\quad$ $\mathbf{k}' = \text{ComputeCenterCoordinates}(\mathbf{c})$
7: $\quad$ $d_{min} = \infty$
8: $\quad$ **for** $\mathbf{k} \in \mathbb{K}_{gt}$ **do**
9: $\quad\quad$ $d = \text{ComputeEuclideanDistance}(\mathbf{k}', \mathbf{k})$
10: $\quad\quad$ **if** $d < d_{min}$ **then**
11: $\quad\quad\quad$ $d_{min} = d$
12: $\quad\quad\quad$ $t = \text{GetKeypointType}(\mathbf{k})$
13: $\quad\quad$ **end if**
14: $\quad$ **end for**
15: $\quad$ **if** $d_{min} < \xi$ **then**
16: $\quad\quad$ **continue**
17: $\quad$ **end if**
18: $\quad$ $\mathbb{K}_{pred} = \mathbb{K}_{pred} \cup \{(\mathbf{k}', t)\}$
19: **end for**
20: **return** $\mathbb{K}_{pred}$

---

### C.1    Post-processing for Keypoints

For keypoints, since all keypoints were labeled in red during training, our first step in post-processing is to extract all red regions from the RGB output. Next, we identify all connected components within the extracted red regions. For each connected component, we further extract sub-regions that approximate a circular shape. This step is crucial because, in some cases, multiple predicted

---

**Algorithm 2** Segmentation Post-processing

---

**Input:** RGB segmentation mask $\mathbf{m}$, RGB tolerance $\sigma$, area threshold $\xi$, kernel size $k$, connected components number threshold $\eta$, duplicate mask threshold $\beta$

**Output:** extracted masks $\mathbb{M}_{pred}$

1: Get the number of peaks $p$ of the histogram of $\mathbf{m}$
2: Get the number of clusters $n = Mean(p)$
3: Get the clustered colors by $\mathbb{C} = \text{KMeans}(\mathbf{m}, n)$
4: $\mathbb{M}_{pred} = \varnothing$
5: **for** $c \in \mathbb{C}$ **do**
6:     **if** IsCloseToBlack$(c, \sigma)$ **then**
7:         **continue**
8:     **end if**
9:     $\mathbf{m}' = \text{GetMaskByRGB}(\mathbf{m}, c, \sigma)$
10:    $\mathbf{m}' = \text{BinaryFillHoles}(\mathbf{m}')$
11:    $\mathbf{m}' = \text{RefineWithMorphology}(\mathbf{m}', k)$
12:    $a = \text{GetArea}(\mathbf{m}')$
13:    **if** $a < \xi$ **then**
14:       **continue**
15:    **end if**
16:    $y = \text{GetConnectedComponentsNumber}(\mathbf{m}')$
17:    **if** $y > \eta$ **then**
18:       **continue**
19:    **end if**
20:    $\mathbb{M}_{pred} = \mathbb{M}_{pred} \cup \{\mathbf{m}'\}$
21: **end for**
22: $\mathbb{M}_{pred} = \text{RemoveDuplicateMasks}(\mathbb{M}_{pred}, \beta)$
23: **return** $\mathbb{M}_{pred}$

---

keypoints may overlap, requiring us to separate them as much as possible. For example, when a person clasps his hands together, the keypoints for both hands may overlap.

Once the circular regions are identified, we compute their center points as the predicted keypoint coordinates. Since our model does not explicitly predict the type of each keypoint (*e.g.*, hand, foot), we assign keypoint types by measuring the distance between the extracted keypoints and the ground-truth (GT) keypoints. Each predicted keypoint is assigned the type of its nearest GT keypoint. To ensure robustness, we apply a distance threshold, considering only those predicted keypoints that are sufficiently close to a GT keypoint. Finally, all extracted keypoints that are successfully matched to a GT keypoint form our final predicted keypoint coordinates after post-processing. The algorithm is shown in Algorithm 1.

## C.2 Post-processing for RGB Masks

For entity segmentation and instance segmentation RGB masks, we employ clustering algorithms to extract the object masks. Specifically, we first compute the histogram peaks for each of the three RGB channels and estimate the number of clusters by averaging the peak counts across the three channels. We then use KMeans clustering to group the colors and identify the clustered regions in the RGB mask. For each identified cluster, we extract regions with RGB values close to the cluster's centroid. This step is followed by morphological operations to refine the extracted masks, such as filling holes and removing small, fragmented regions. We further filter the masks by computing their area, excluding any regions that are too small to be meaningful.

Table S13: When post-processing RGB masks, small regions and excessive numbers of objects significantly lead to performance degradation.

| Category | AP $\uparrow$ |
|---|---|
| Bear | 76.3 |
| Dog | 68.9 |
| Cat | 71.7 |
| Person | 18.6 |
| Bird | 10.4 |
| Book | 10.8 |

Additionally, we also consider the number of connected components within the extracted masks, discarding overly fragmented results that have too many connected components. Finally, we refine the extracted masks by calculating the Intersection over Union (IoU) between them, removing any duplicate or overlapping masks. The algorithm is shown in Algorithm 2.

### C.3   Performance Degradation of Keypoints

For human keypoints, the Performance degradation is primarily due to two factors: Firstly, we utilize skeletal-form RGB images rather than heatmaps. While the former produces visually appealing results, the extraction of keypoints during post-processing introduces considerable errors. Secondly, our evaluation follows the $192\times256$ top-down human keypoints protocol. The original $192\times256$ images are resized to $768\times768$ before being input into the model, resulting in extremely blurred inputs that likely contribute to the diminished performance.

### C.4   Performance Degradation of RGB Masks

We observe that while the quality of our instance segmentation visualizations is high, the average precision (AP) for certain categories remains unsatisfactory. For example, for the Person category, we conducted exhaustive experiments and achieved good visualization results (highlighted by the green rectangle in Figure S9), but AP is low (as in Table S13).

We trace the root cause of metrics degradation during post-processing and find that this is particularly due to small objects and an excessive number of objects. Specifically, during mask processing, we filter out small noise regions. The genesis of these artifacts is predominantly attributed to subtle colorimetric fluctuations or minor inconsistencies in pixel values within areas of a mask intended to be uniformly colored. However, this operation also removes some positive samples, such as the crowd and the bird highlighted in red in rows 3 to 5 in Figure S9. These samples are susceptible to being misidentified as noise due to their diminutive size. Despite this limitation, the filtering of these noise regions is maintained because their persistence would otherwise exert a more detrimental impact on the quality of the final results. In our setting, filtering noise regions results in better metrics compared to not filtering them. Additionally, when an image contains an excessive number of objects of the same category (as in row 6 of Figure S9), post-processing may erroneously group similarly colored but distinct objects into a single class, leading to lower metrics. Furthermore, as in Table S13, we examine categories with fewer small objects and instances of those categories, such as bear, dog, and cat, and observe higher AP scores. However, for categories with opposite characteristics, their AP scores tend to be lower. This phenomenon is also observed in entity segmentation, which further elucidates why our entity segmentation results exhibit lower scores on small objects.

Although we can optimize post-processing by adjusting hyperparameters for each image to achieve the best results, this approach becomes impractical for large-scale evaluation, as it requires significant manual effort. Consequently, the dependency on post-processing remains a limitation of our approach.

## D   Additional Results

### D.1   Additional Visualizations

We present additional visualization results of our method across various tasks, as can be seen in Figures S8, S11, S10, S15, S16, S17, S18, S19, S20, S21, S22. For interactive segmentation, we compare our approach with SAM. These results strongly demonstrate the potential of DICEPTION. DICEPTION is capable of achieving high-quality results, even in challenging scenarios. Furthermore, the few-shot fine-tuning of DICEPTION, which requires minimal data and trainable parameters, strongly demonstrates the remarkable transferability of DICEPTION to tackle new tasks. Our DICEPTION is capable of further refining the segmentation of fine details, such as intricate hair structures, achieving matting-level performance.

### D.2   Comparative Experiments with One Diffusion

Qualitative visual comparisons between our method and One Diffusion in Figure S14 highlight key distinctions. In segmentation, our approach excels by simultaneously segmenting objects by semantic

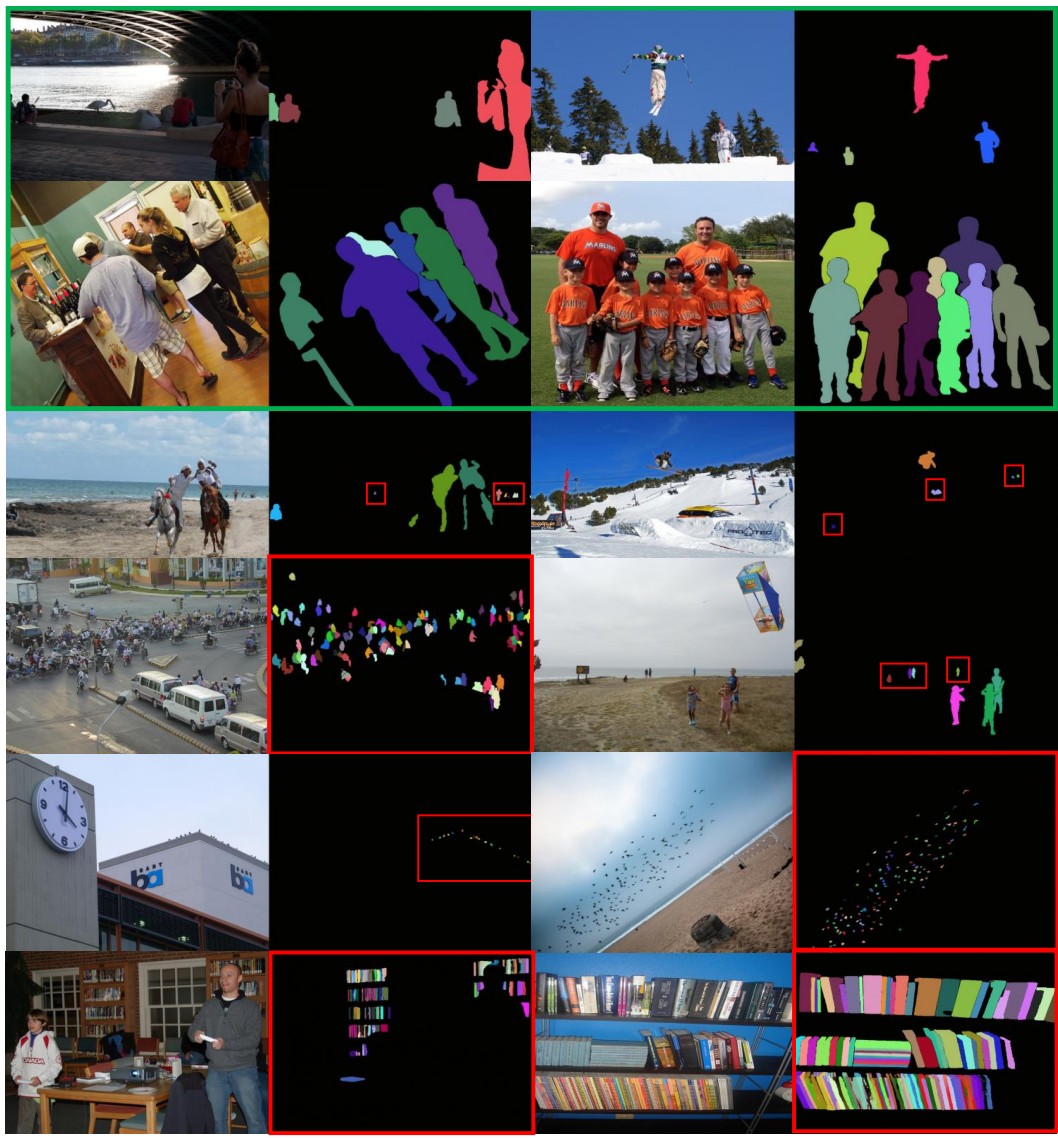

Figure S9: When post-processing RGB masks, small regions and excessive numbers of objects lead to significant metric degradation.

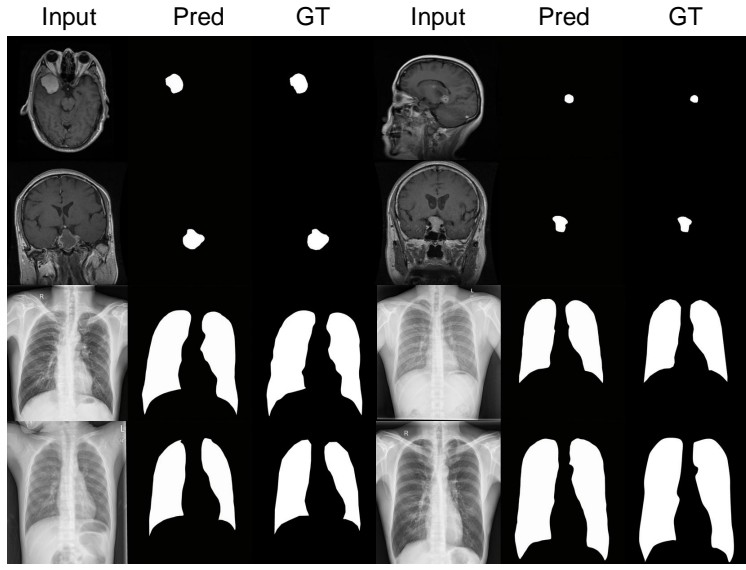

Figure S10: Additional few-shot fine-tuning results on lung segmentation and tumor segmentation.

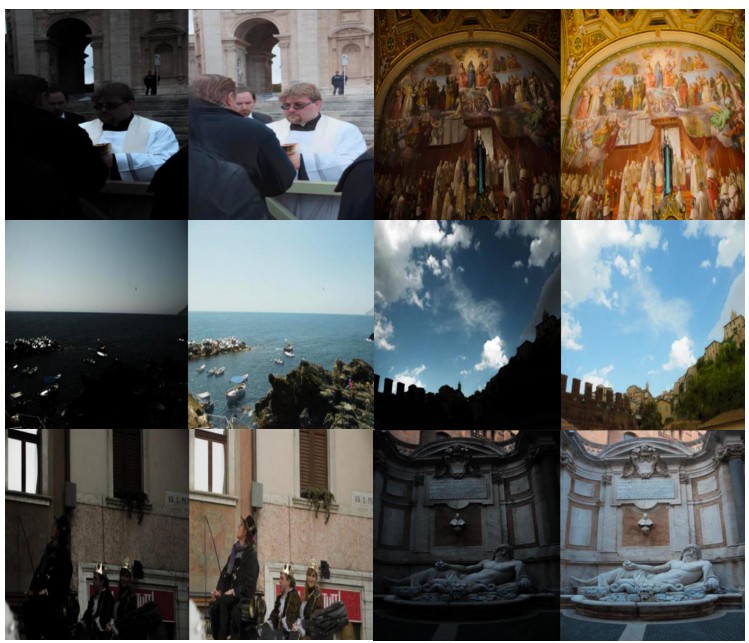

Figure S11: Additional few-shot fine-tuning results on image highlighting.

class and differentiating individual instances—a capability lacking in One Diffusion. Moreover, the segmentation quality of our method is superior to that of One Diffusion, especially in object-dense regions where the latter exhibits noticeable performance degradation.

A critical limitation of One Diffusion is its apparent inability to distinguish input images from conditioning signals, leading to a conflation of image understanding tasks with image generation. For example, when performing human keypoint estimation, One Diffusion may erroneously generate an image depicting a similar pose rather than predicting the actual keypoints. Conversely, our model, being fundamentally oriented towards image perception, not only consistently yields high-quality, accurate results without confusion, but also performs challenging perception tasks inaccessible to One Diffusion, such as interactive segmentation.

## E    Discussions and Limitations

**Discussions**    Our method highlights the following key findings:

- The inherent prior knowledge of diffusion models is highly effective for perception tasks. By leveraging this prior effectively, our approach enables a single model to address multiple tasks. Notably, it achieves performance comparable to existing single-task specialized models, even on challenging tasks such as interactive segmentation, and does so with limited data.
- Our comprehensive experimental evaluation demonstrates that token-wise concatenation is the most efficient and effective strategy for leveraging the prior knowledge of transformer-based diffusion models. Furthermore, we provide evidence that the DiT architecture works better compared to U-Net. This is attributed to the fact that transferring U-Net to multiple perception tasks not only introduces additional parameters that can potentially disrupt the pre-trained model's prior but also suffers from significant information loss due to its inherent downsampling operations.
- A modest CFG value can yield performance improvements for pixel-sensitive tasks such as depth and normal estimation.
- We find that flow-matching models, when trained in a multi-step denoising setting, naturally support few-step inference for perception tasks.
- Our DICEPTION exhibits a faster and more effective adaptation capability to new downstream tasks.
- The efficacy of our approach is demonstrated on a different DiT architecture and smaller model, indicating its robustness.
- The model demonstrates strong capability of detail preservation after pixel-aligned training on perception tasks.

To the best of our knowledge, we are the first to successfully leverage diffusion priors to address multiple perception tasks with a single model without exceptionally large or cherry-picking high-quality data, achieving performance on par with specialized models, even on the challenging interactive segmentation compared with SAM. **In our view, the capabilities of our method are far from being fully realized**, and further training with larger, higher-quality datasets has the potential to yield even more compelling results. For instance, in high-level tasks such as referring segmentation shown in Figure S12, our model achieves results with finer details than the ground truth. This not only demonstrates the model's ability to benefit from related tasks but also showcases its strong semantic understanding. Furthermore, we observe early signs of task composition in our model, albeit with a low success rate. For instance, the model can predict the depth or normal map of an object indicated by point inputs while generating a black mask for other regions, as illustrated in Figure S13, though the success rate is very low. In conclusion, we believe that our work not only presents a generalist model with a vast capacity for improvement, but also provides comprehensive experiments and analyses that can serve as a valuable foundation for future research.

**Limitations**    Although our DICEPTION achieves great results across multiple tasks, our model, as a diffusion model, leads to relatively longer inference times. On one H800, it takes an average of 0.8 seconds to process a single image. On one 4090-GPU card, inference for one image takes approximately 2 seconds. We believe that this issue can be addressed through few-step diffusion techniques, which we leave for future works.

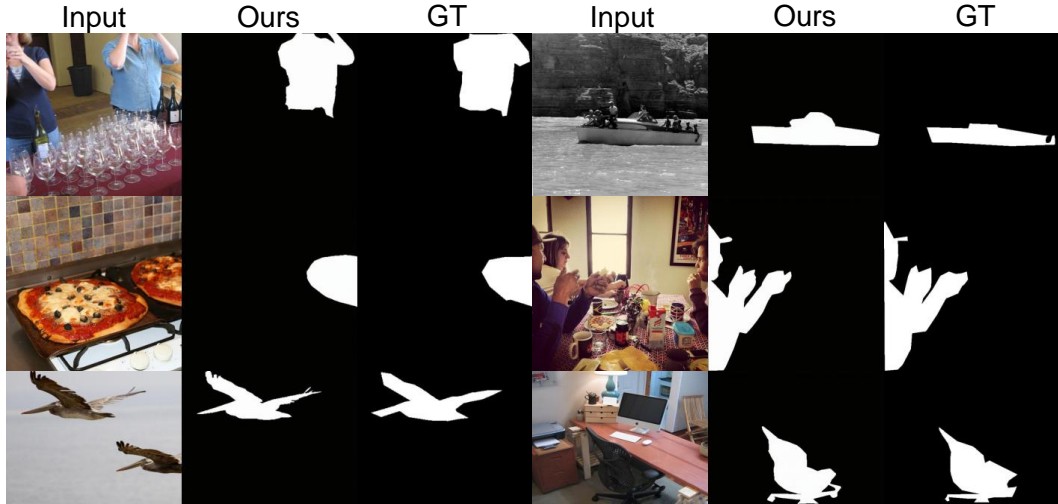

Figure S12: DICEPTION achieves finer results on referring segmentation, showing the potential of mutual improvement between related tasks.

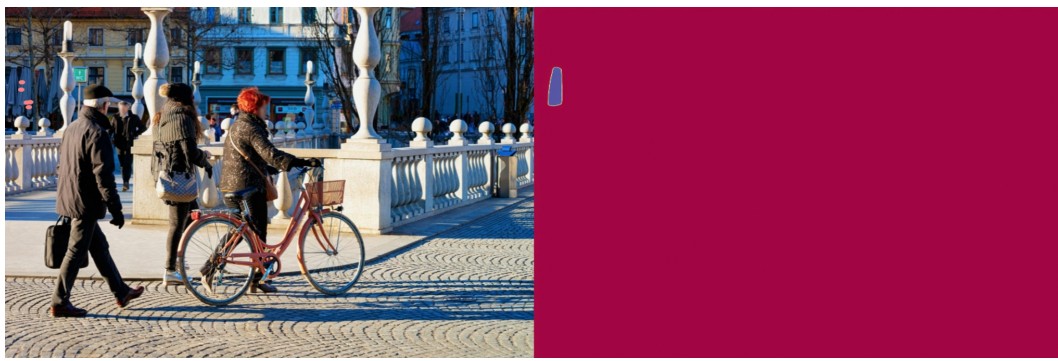

Figure S13: Example of task composition. Our model can isolate a point-specified object to generate its corresponding depth map, while correctly suppressing predictions for all other regions. Although the success rate is very low, this result still reveals a promising capability.

Furthermore, our evaluation on certain tasks such as human keypoints estimation and text-based instance segmentation necessitates post-processing, which can introduce substantial errors. However, unlike some contemporary diffusion-based works [55, 118] that often omit quantitative evaluation on the task such as human keypoints estimation, we take a step further by providing evaluation metrics. Our analysis demonstrates that lower scores on these tasks are not due to model performance but are significantly influenced by the post-processing step. Consequently, the dependence on post-processing for quantitative evaluation on certain tasks remains a limitation of our method. Despite the limitations, we believe that DICEPTION is a valuable exploration for diffusion-based generalist visual perception foundation models.

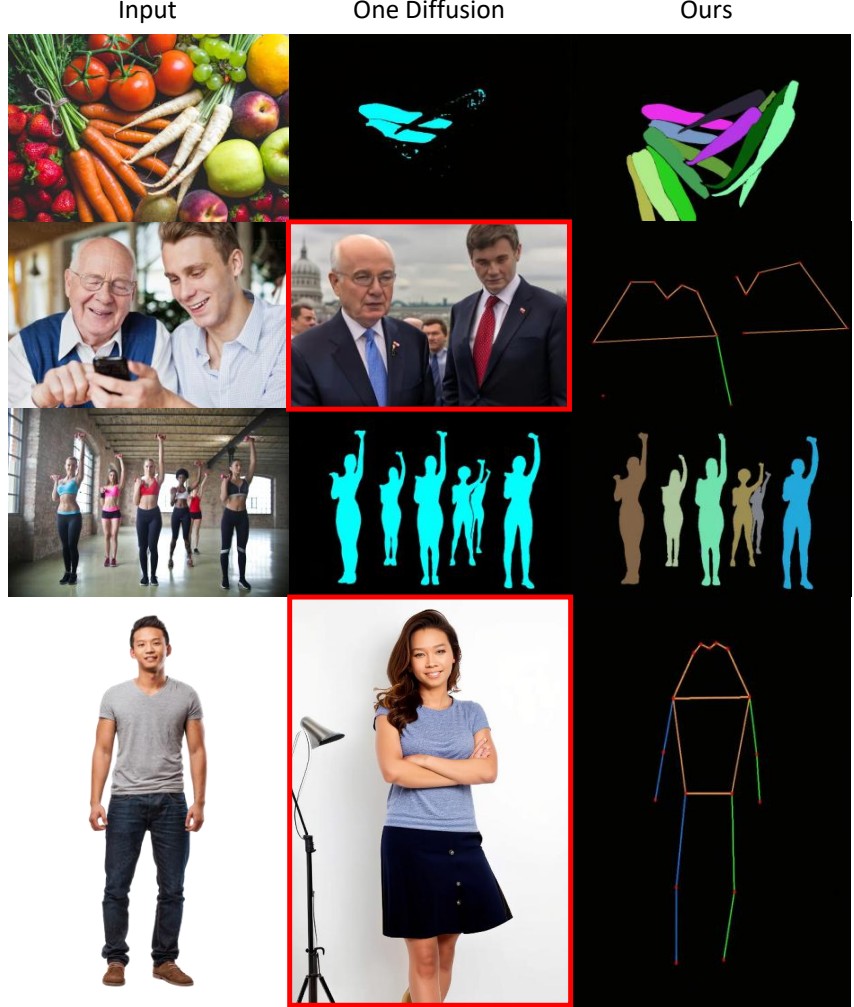

Figure S14: Our segmentation not only separates semantically identical objects but also distinguishes different instances of the same category, achieving higher segmentation quality. Moreover, One Diffusion tends to generate an image similar to the input when performing image understanding tasks, as red-highlighted in the figure.

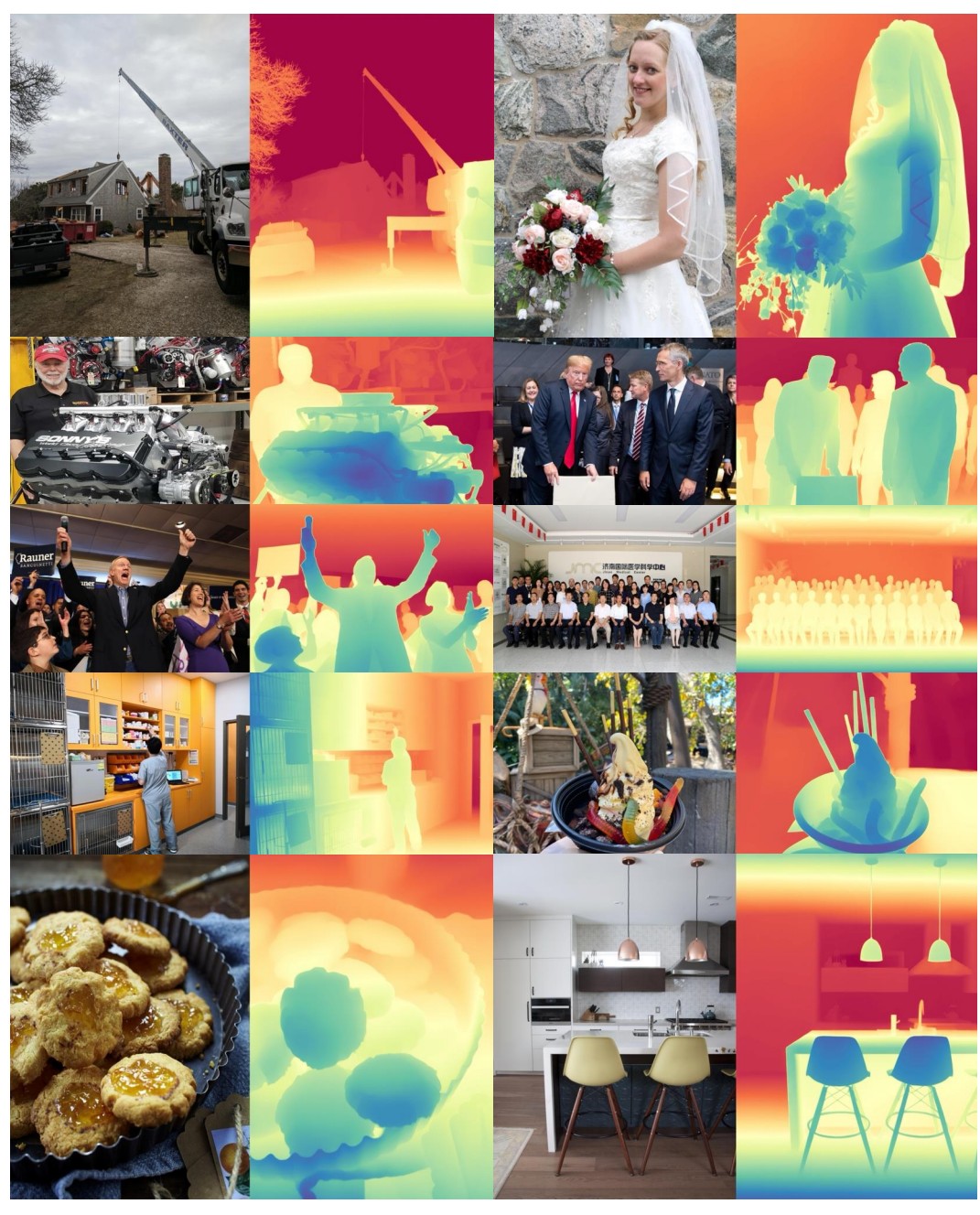

Figure S15:  Additional depth estimation visualizations.

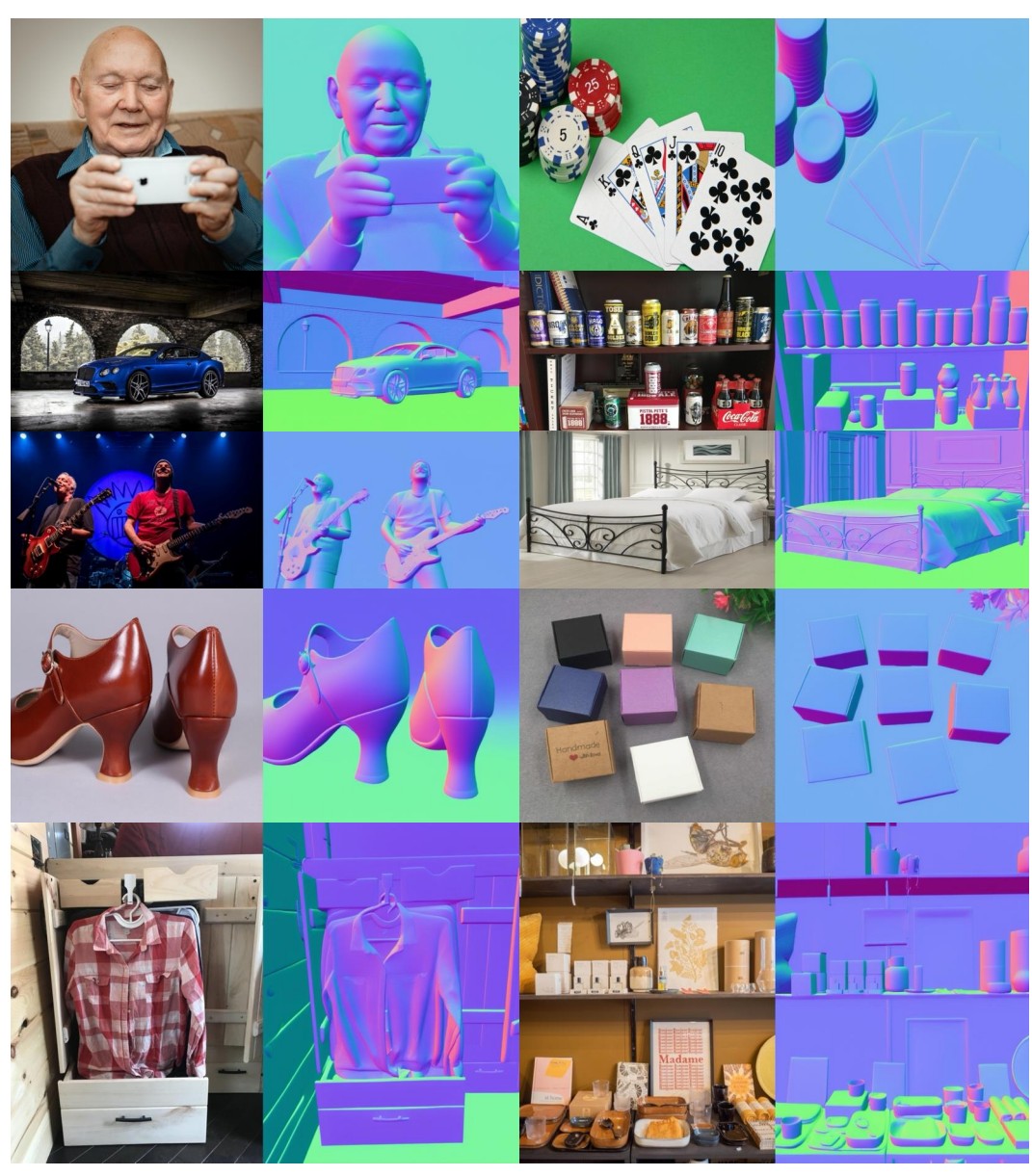

Figure S16: Additional normal visualizations.

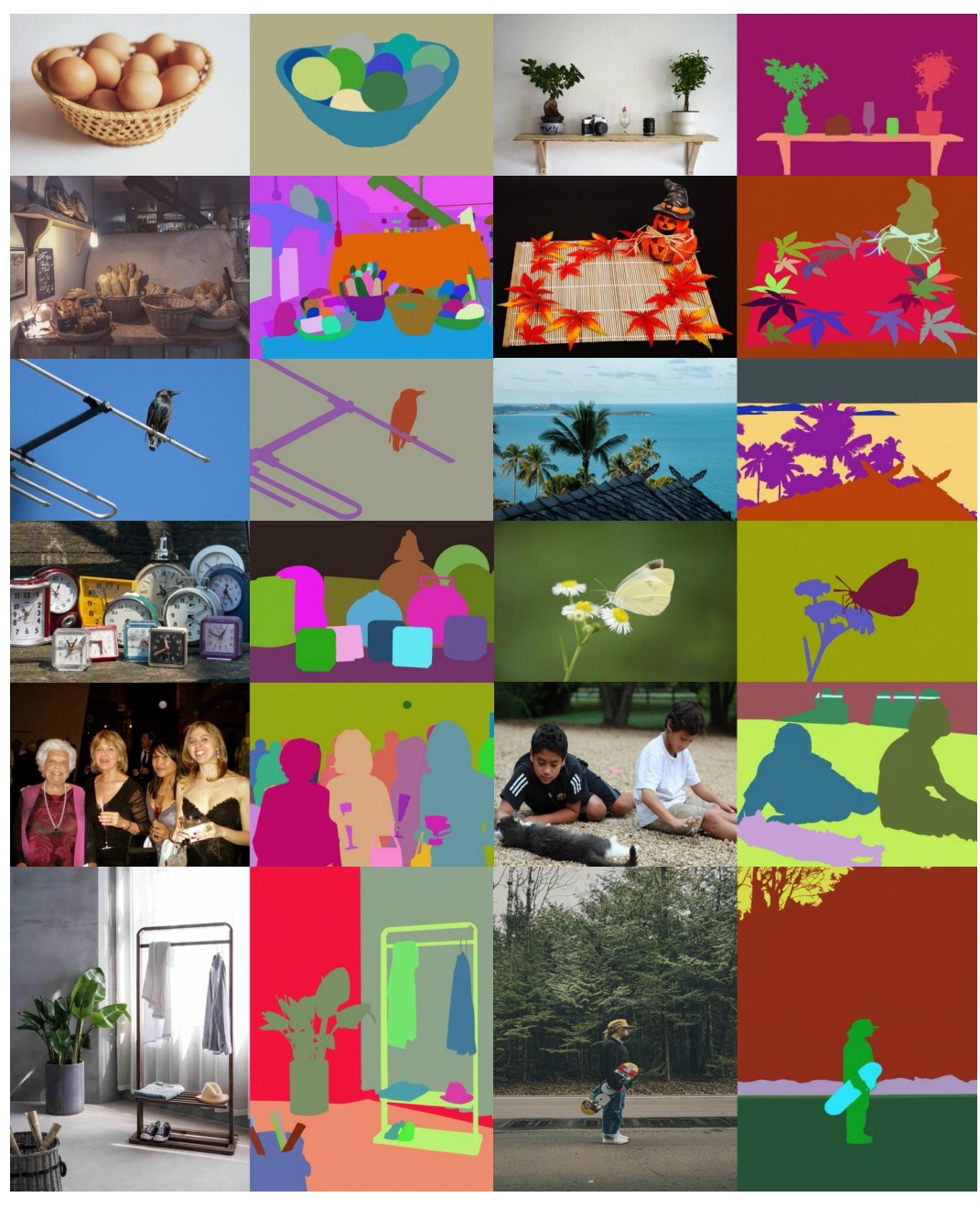

Figure S17: Additional entity segmentation visualizations.

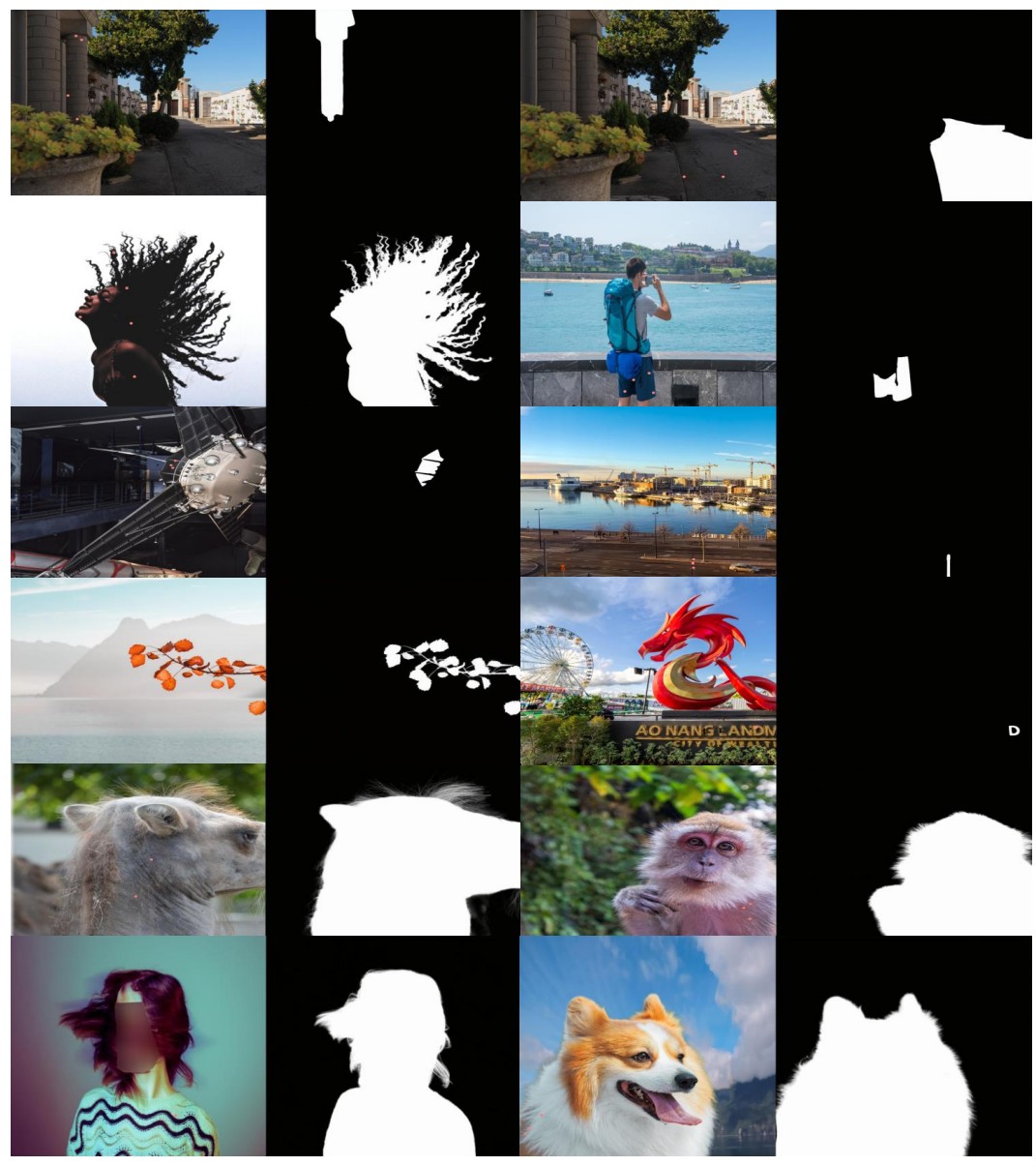

Figure S18: Additional interactive segmentation visualizations.

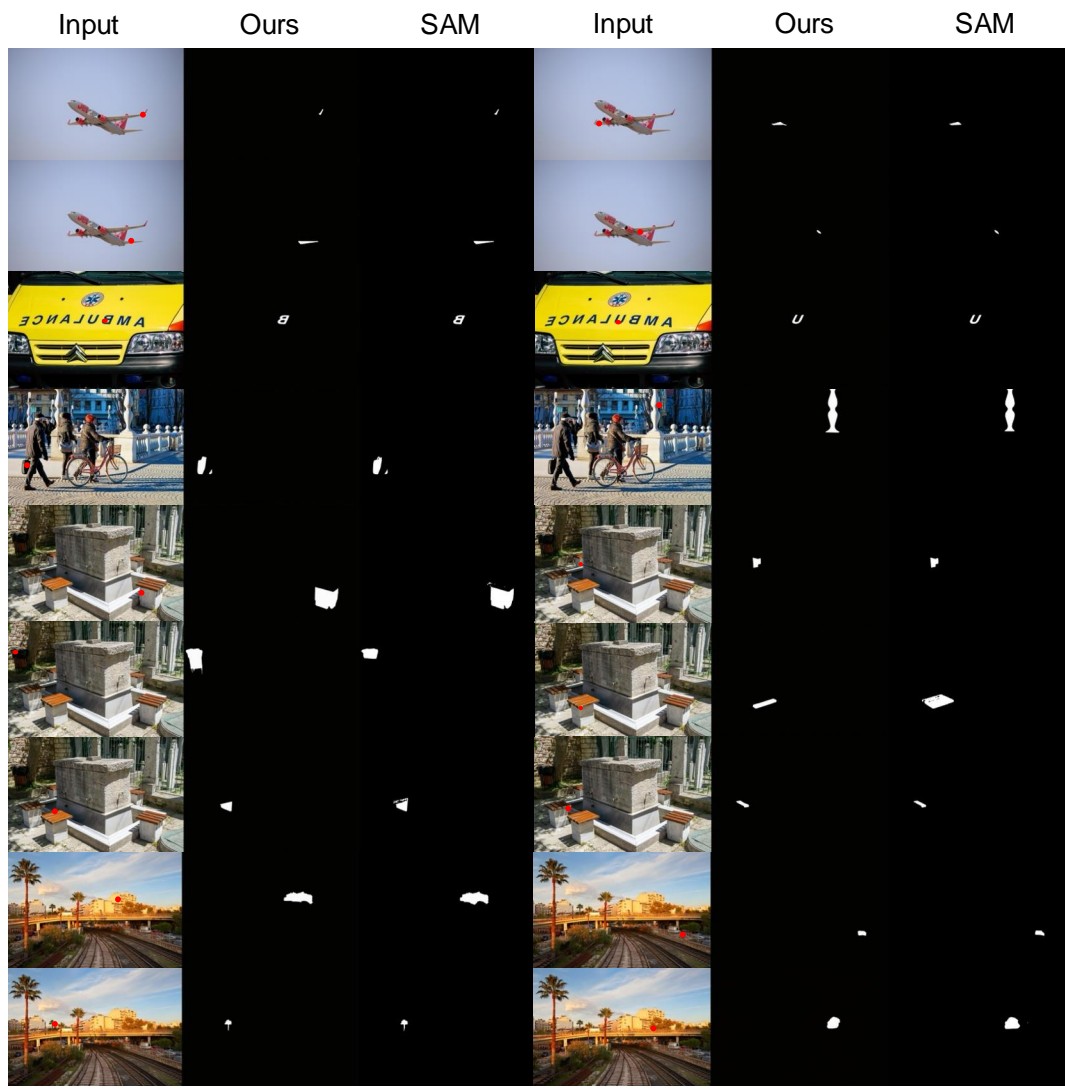

Figure S19: Comparison of the segmentation results between DICEPTION and SAM-vit-h with 1-point input.

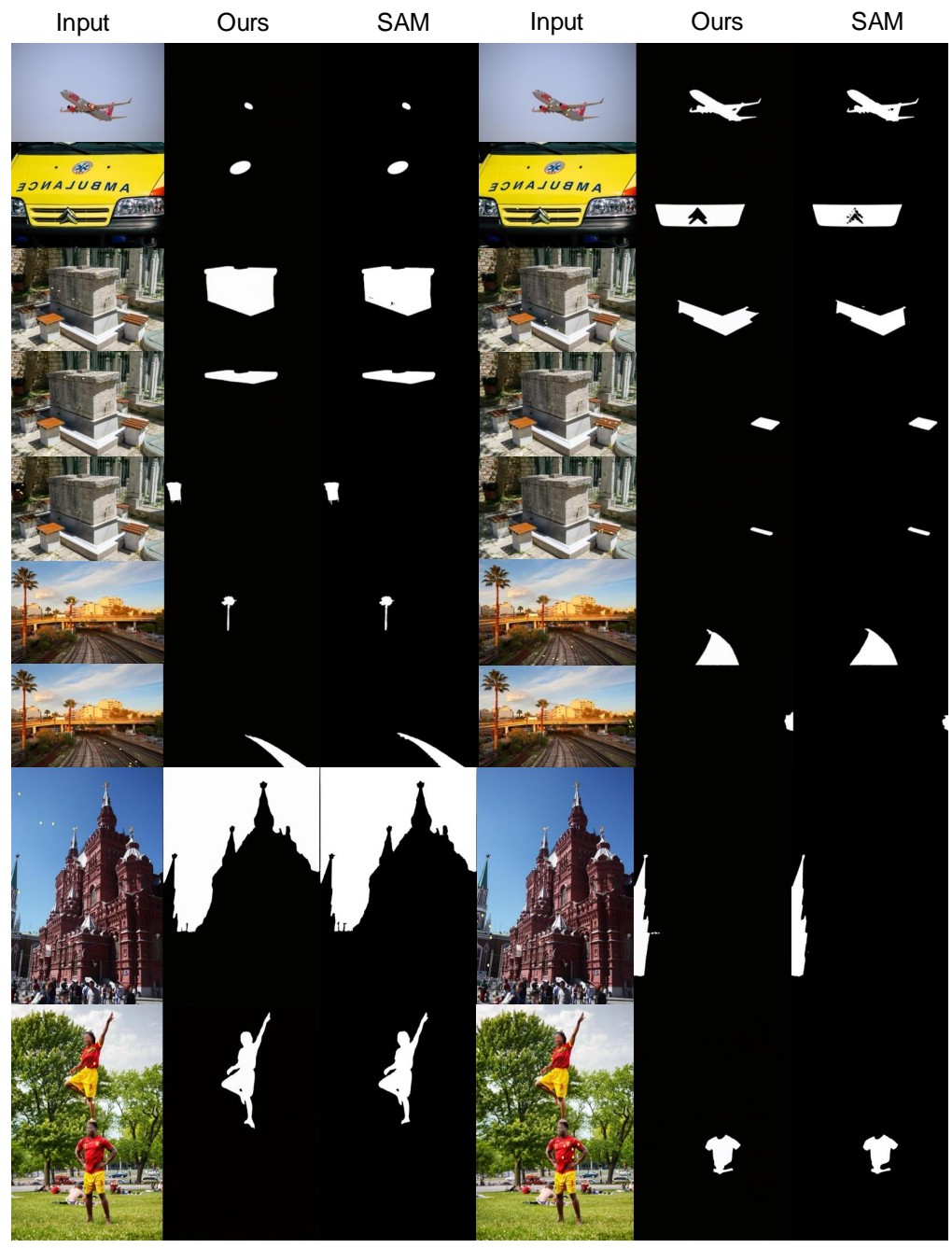

Figure S20: Comparison of the segmentation results between DICEPTION and SAM-vit-h with 5-point input.

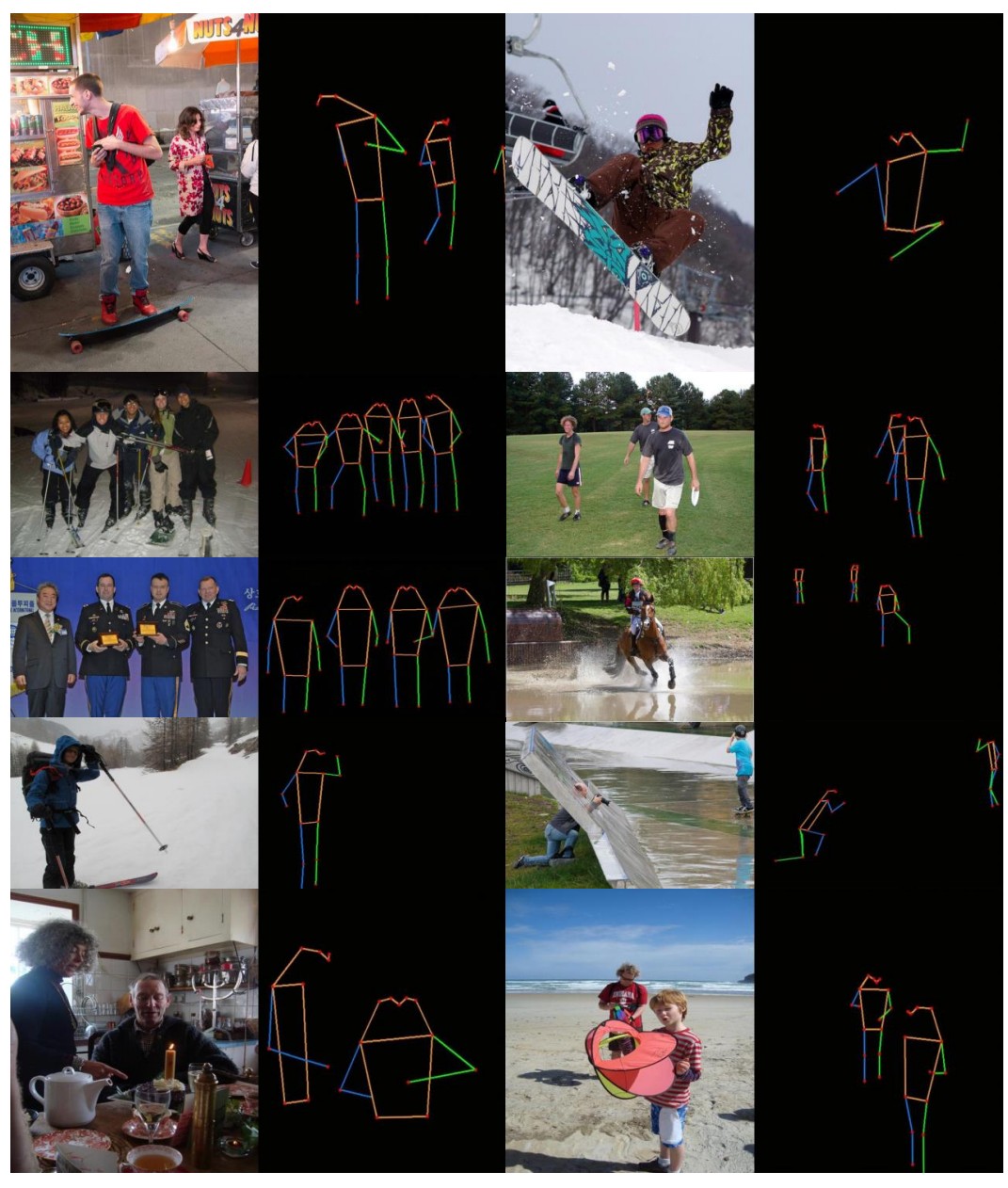

Figure S21: Additional pose estimation visualizations.

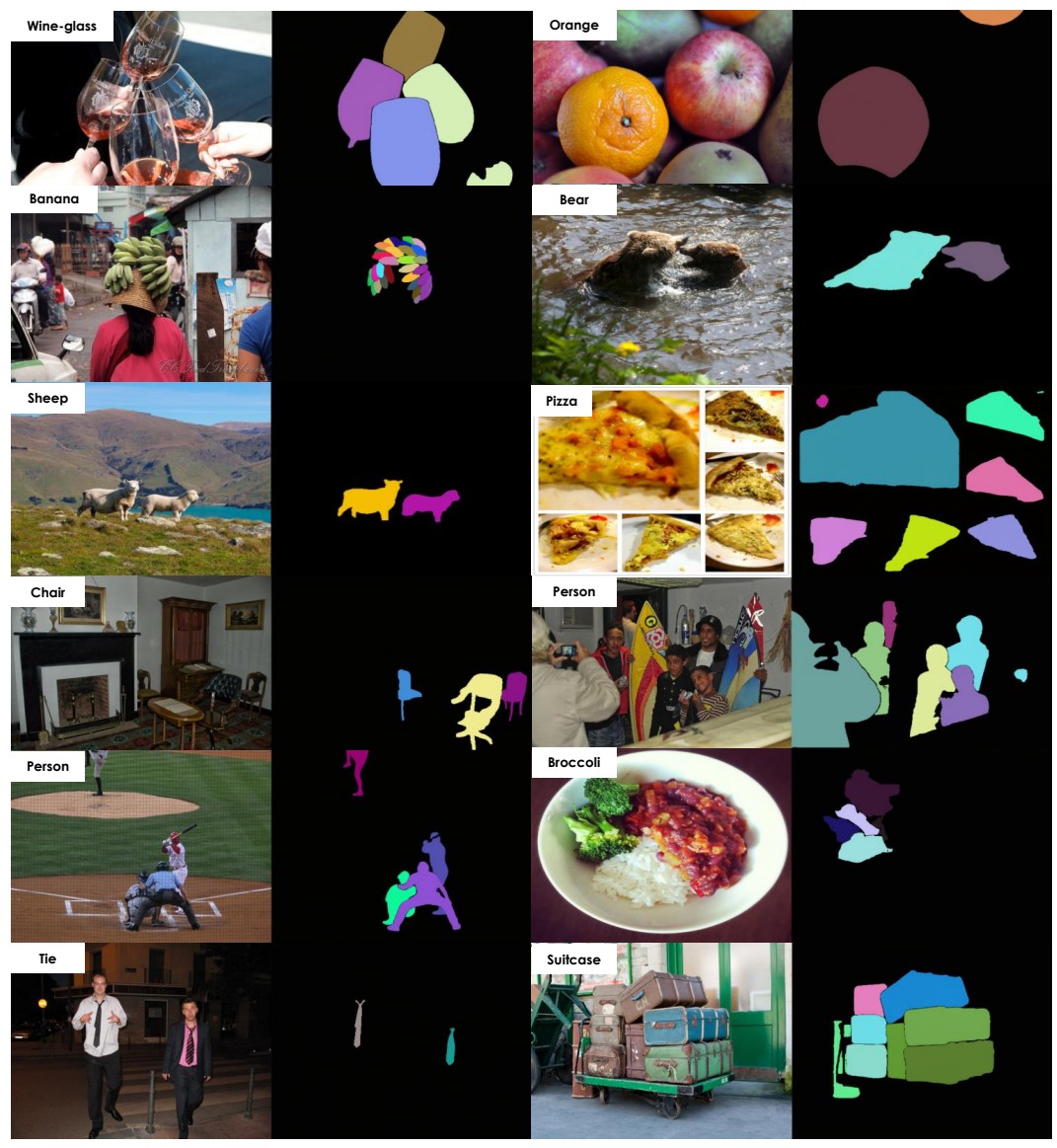

Figure S22: Additional text-based instance segmentation visualizations.

