# OpenReview forum: "DICEPTION: A Generalist Diffusion Model for Visual Perceptual Tasks"
_NeurIPS.cc/2025/Conference — NeurIPS 2025 spotlight_

### Official Review · Reviewer_Taie · 2025-06-10

**Clarity:** 3
**Significance:** 3
**Originality:** 2
**Rating:** 5
**Confidence:** 5

**Summary:**

This paper proposes a perception generalist built from diffusion models that could handle multiple tasks simultaneously. The major recipe is to collect the diverse perception training data, train them at scale (it is really brave to train for 24 days), and prove that such a paradigm could lead to strong models. The performance suggests that this framework can address a wide range of perception tasks.

**Questions:**

Another follow-up question:

* It could be a vague question, but it would definitely be meaningful to the field. Have the authors considered any type of "emergent behaviors"? For example, can the model compose two tasks together just by simply composing the prompts, e.g., draw the depth map while outputting some segmentation masks; or if the model can do things in multi-turn, such as sequentially conducting multiple tasks one by one?

**Ethical Concerns:**

["NO or VERY MINOR ethics concerns only"]

**Final Justification:**

The additional experiments have made this paper a significant improvement over the previous diffusion-based perception study.

**Limitations:**

Yes, the authors mention that no societal impact.

**Paper Formatting Concerns:**

No formatting issues

**Quality:**

3

**Strengths And Weaknesses:**

## Strength
* I think developing a generalist perception model is a crucial topic, and this paper presents a valuable exploration with diffusion models.
* The evaluation and training at scale make the efforts in the paper inspiring to the researchers following this direction.

## Weakness

Perhaps this paper would be judged from the aspect of lacking "novelty," but I tend to think "novelty" is not a good criterion for this work. As a researcher who has worked in taming diffusion models for perception, I understand the difficulties of re-purposing diffusion models for perception tasks. That said, I would mainly list the weakness based on **what are the critical problems in diffusion for perception that have not been addressed by this paper**, focusing on whether this paper brings new knowledge to people.

1. To begin with, more context for the authors: some previous papers applying diffusion for perception or addressing such problems should be mentioned and discussed. The authors seem not to cover (all) of them.

a. Emu Edit: Precise Image Editing via Recognition and Generation Tasks. CVPR 2024.

b. InstructCV: Instruction-tuned text-to-image diffusion models as vision generalists. ICLR 2024.

c. ADDP: Aligning Generative Denoising with Discriminative Objectives Unleashes Diffusion for Visual Perception. ICLR 2025.

d. Instructdiffusion: A generalist modeling interface for vision tasks. CVPR 2024.

2. Something the previous papers have demonstrated is that: diffusion-based models can already perform well for tasks like depth estimation and normal estimation (such as Marigold), but are fundamentally worse on tasks requiring detailed understanding and multi-modal understanding. Under such a context, this paper also performs badly on these kinds of challenging scenarios (Table 3, Table 4, Table 6), leading to the problems of:

* The results shown in this paper do not fundamentally change the picture of diffusion for perception, which limits my rating for this paper. My borderline acceptance is mainly appreciating the author's exploration on the depth estimation and normal estimation part, because I understand how hard it is to make diffusion models work on such instance-level fine-grained perception tasks.

* The authors also dangerously over-claim that "In this paper, we successfully leverage the priors of diffusion models to achieve results on par with the state-of-the-art models on various tasks with only minimal training data", which seems not to be true and leave the readers with wrong impressions.

3. That being said, I would like to significantly increase the score if the authors can show good performance on some multi-modal or instance-level perception task. Since the backbone is SD3, with a strong text encoder of T5, would the authors consider referring segmentation for a try? From the previous papers of InstructDiffusion and ADDP, this seems a more viable task for diffusion-based perception.

---

> ### Author Rebuttal · Authors · 2025-07-30
>
> We sincerely thank the reviewer for the professional, thoughtful and constructive comments!
>
> ### **Weaknesses**
> **W.1 Missing Some Related Works**
>
> We appreciate the reviewer’s professional suggestion to strengthen the contextual grounding of our contributions. We will discuss and analyze these works in the revised paper.
>
> **W.2 Not Good Performance on Certain Tasks**
>
> Thank you for your insightful comment! We respectfully believe the concern is not due to limitations in model's performance, but rather due to the lack of a standardized output representation that is both aligned with the diffusion model's predictions and suitable for established evaluation metrics. Owing to the significant variation in representations among different tasks, existing methods typically employ task-specific decoders, especially for instance-level tasks such as pose estimation.
>
> Unlike prior works that often omit quantitative evaluation on these hard-to-evaluate perceptual tasks, our paper provides usable quantitative metrics. We believe this is a critical issue that cannot be overlooked. As diffusion models for perception continue to develop, establishing ways to compute such metrics will be essential for fair and rigorous comparison across methods. Relying only on easy-to-evaluate tasks such as depth or surface normal estimation is insufficient to fully distinguish the performance of different approaches. Although our current metrics—affected by significant post-processing errors—may appear to underperform compared to traditional task-specific methods, we believe that they can still serve as a practical and necessary evaluation baseline for future diffusion-based perceptual models.
>
> Furthermore, we believe these challenges can be addressed. For instance, by representing poses with heatmaps, although the errors in post-processing still exists, we achieve notably higher evaluation metrics. Furthermore, we think that a more fundamental solution may lies in exploring the VAE decoder. Existing work[1] shows fine-tuning VAE decoders has yielded higher-quality point maps. We believe this is insightful and promising.
>
> [1] GeometryCrafter: Consistent Geometry Estimation for Open-world Videos with Diffusion Priors
>
> | | HRNet | HRFormer | ViTPose | Painter | Ours | Ours-Heatmap |
> |:-:|:--:|:--:|:--:|:--:|:--:|:---:|
> | **AP** | 76.3| 77.2 |78.3| 72.5| 57.8 | 68.9 |
>
> **W.3 Key challenges in Diffusion Model for Perception**
>
> Thank you for your professional comments! In our opinion, **inference speed** is one of the critical problems when applying diffusion models to perception tasks, due to the typical requirement for many iterative steps. Although previous works have achieved 1-step inference in single-task settings, this problem is still under-explored in multi-task scenarios.
>
> We explore the feasibility of few-step even 1-step inference and find that **our model demonstrates high-quality results in few-step inference settings without significant performance drop.** Different tasks exhibit varying sensitivities to the reduction of inference steps. For instance, depth and normal estimation can be performed with as few as one inference step without significant performance degradation. For more intricate tasks like interactive segmentation, while a single step leads to degradation, a moderate increase to 7 inference steps still yields results with minimal performance compromise, which is a 4x acceleration. **To the best of our knowledge, this is the first time such a capability is demonstrated in diffusion model for multi-task perception.** It strongly supports the advantage of flow-matching-based diffusion models in solving perception tasks.
>
> | Depth | KITTI | | NYUv2| | ScanNet |  |DIODE | | ETH3D  |  |
> |:--|:--|:--|:--|:--|:--|:--|:--|:--|:--|:--|
> |  | absrel↓ | delta_1↑ | absrel↓ | delta_1↑ | absrel↓ | delta_1↑ | absrel↓ | delta_1↑ | absrel↓ | delta_1↑ |
> | 28-step | 0.069 | 0.949 | 0.061| 0.960 | 0.072 | 0.944 | 0.289  | 0.722  | 0.050 | 0.975 |
> | 14-step | 0.077 | 0.942 | 0.063 | 0.958| 0.074| 0.943 | 0.272  | 0.718  | 0.048 | 0.978|
> | 7-step | 0.081 | 0.939 | 0.065| 0.953 | 0.078| 0.943 | 0.286  | 0.714  | 0.052 | 0.971|
> | 3-step | 0.083 | 0.938 | 0.069 | 0.953 | 0.077| 0.940| 0.294  | 0.707  | 0.063 | 0.967|
> | 1-step | 0.086 | 0.936 | 0.072| 0.945  | 0.076| 0.937| 0.305  | 0.702  | 0.065 | 0.967|
>
> | Normal |  |  | nyu|   |    |
> |:--|:--|:--|:--|:--|:--|
> | | mean ↓ | med ↓ | 11.25 ↑ | 22.5 ↑| 30 ↑|
> | 28-step | 18.338| 10.106| 52.850 | 77.079 | 82.903 |
> | 14-step | 18.631| 10.463 | 52.837 | 75.288 | 81.682 |
> | 7-step  | 18.335| 10.492 | 52.771 | 75.443 | 81.936 |
> | 3-step  | 18.067| 10.417 | 53.046 | 76.500 | 81.673 |
> | 1-step  | 18.094| 10.382 | 51.839 | 76.575 | 81.371 |
>
> | Interactive Segmentation | 28-step  | 14-step  | 7-step | 3-step | 1-step  |
> |:--|:--|:--|:--|:--|:--|
> | mIoU of 23 validation datasets | 47.10| 47.01| 46.89 | 45.18 | 42.53 |
>
> We believe this is because flow matching explicitly imposes linear constraints on intermediate noisy latents, enforcing them to be linear interpolations between noise and the target. As a result, the denoising trajectory is encouraged to be as straight as possible. Moreover, low-level perception tasks typically do not require conditioning on complex prompts. Together, these factors enable the model to produce accurate outputs even with a reduced number of inference steps, as the inference trajectory remains well-aligned with the learned straight path.
>
> We also apologize for the confusing presentation in Section 4.3. In Section 4.3 of our paper, we follow the GenPercept training strategy, where the model is directly trained to **predict the target from pure noise in a single step.** This approach discards the linear constraints on intermediate noisy latents that are explicitly enforced by flow matching, thus producing poor results.  Here we show that **when the model is trained under a multi-step flow matching denoising setting, it can naturally support few-step inference.** We will revise this part of the paper accordingly.
>
> In summary, **our paper contributes new knowledge to the community by showing that flow-matching-based diffusion models, when trained with multi-step denoising, inherently support efficient few-step and even one-step inference in perception tasks.**
> We kindly refer you to our response to Reviewer MKSc, where we provide more comprehensive analysis and detailed experimental results.
>
> **W.4 Dangerous Expressions**
>
> Thank you for pointing this out. What we intended to convey is that, our approach can achieve competitive performance using substantially less training data compared to specialized expert models (our 600K images vs. 1B for SAM,  500K vs. 62M samples for DepthAnything2). We will revise the wording for better clarity.
>
> **W.5 Referring Segmentation**
>
> Thank you for your professional suggestions! We conduct additional referring segmentation experiments using RefCOCOg and demonstrate our method can show good performance. We will also supplement qualitative visualizations in the revised version of the paper, as images cannot be provided in response.
>
> |    | RefCOCOg-val | RefCOCOg-test |
> |:---:|:----:|:----:|
> | LISA-7B  | 67.9   | 70.6 |
> | PixelLM-7B | 69.3  | 70.5 |
> | Ours     | **69.9**   | **71.2**   |
>
> ### **Questions**
>
> **Q.1 Emergent Behaviors and Task Composition**
>
> In our view, emergence behavior typically requires training on extremely large-scale data. The amount of data used in our work is still far from that scale. **Nevertheless, we do observe early signs of such potential in our model. Our model does exhibit certain compositional capability.** Here are some cases:
>
> - When composing segmentation and human pose captions, the model sometimes outputs results with both segmentation masks and pose keypoints. Similar behavior is also observed in depth and surface normal estimation. The model can predict depth or normal of the object labelled by point inputs and output pure black mask for other regions, but with a lower success rate.
> - In entity segmentation, we can provide points as additional input to guide finer-grained predictions. E.g., a bookshelf is segmented as a single entity, but with additional points, the model can further segment the individual books on the shelf.  To summarize, while the model was not trained on combined entity and interactive segmentation data, their integration is achievable at inference through prompt composition.
>
> We will further provide qualitative visualizations above in the revised paper, as images cannot be provided during response.
>
> **As for sequentially conducting multiple tasks one by one, this is fully available. Furthermore, thanks to our model’s few-step inference capability, this process remains highly efficient.**
>
> ### **Discussion**
>
> **D.1 High-level Tasks and Emergent Behaviors**
>
> Tackling complex multi-modal perceptual tasks and exploring "emergent behaviors" are very attractive. In our opinion, achieving these goals requires great understanding capability. In this work, we demonstrate that generative models can be effectively leveraged to build a unified model for multiple perceptual tasks. Recently, a new model which exhibits strong capabilities in both generation and understanding, BAGEL[2], has drawn our attention. It outperforms previous diffusion and vision-language models in both generation and comprehension. We believe this model is promising to solve high-level multi-modal perception tasks. We have made attempts on BAGEL our next step and are currently exploring it. And we are confident this will reveal novel and compelling features on complex, high-level tasks, even emergent behaviors.
>
> [2] BAGEL: Emerging Properties in Unified Multimodal Pretraining
>
> We sincerely thank you again for your professional review, constructive comments, and insightful discussion! Please let us know if you have any further concerns. We also look forward to further constructive discussions with you!  :)

---

> > ### Comment · Reviewer_Taie · 2025-08-03
> >
> > Thanks for the detailed clarification and additional experiments! Your points make sense, but I have several follow-up questions.
> >
> > 1. I don't see sections in the main paper talking about the number of diffusion steps (perhaps I missed it). Could you please clarify what kind of changes in your method enable such improvement in a few-step inference?
> >
> > 2. For the referring segmentation experiments, is this conducted with a VAE decoder or a segmentation decoder?

---

> > > ### Author Response · Authors · 2025-08-03
> > >
> > > Thank you very much for your response!
> > >
> > >
> > > **1 Our model inherently supports few-step inference for perception tasks without any additional techniques.**
> > >
> > > Here is our analysis:
> > >
> > > We attribute this capability to the core mechanism of flow matching, which imposes explicit linear constraints at each intermediate denoising step. In particular, each noisy latent is formulated as a linear interpolation between pure noise and the ground-truth target, which effectively straightens the denoising trajectory. As a result, the model learns to traverse an approximately linear path from noise to the target, even when executed with only a few inference steps.
> > >
> > > We further experiment with PixArt-α, a model based on DiT-style architecture but trained with ODE-based scheduler. When performing inference with fewer steps to address perception tasks such as depth estimation, it experiences a significant drop in performance, further corroborating our analyses regarding the advantages of flow-matching in enabling effective few-step perception inference.
> > >
> > > |         | KITTI   |         | NYUv2   |         | ScanNet |         | DIODE   |         | ETH3D   |         |
> > > |:--------|:--------|:--------|:--------|:--------|:--------|:--------|:--------|:--------|:--------|:--------|
> > > |         | absrel↓  | delta_1↑ | absrel↓  | delta_1↑ | absrel↓  | delta_1↑ | absrel↓  | delta_1↑ | absrel↓  | delta_1↑ |
> > > | 20-step | 0.093   | 0.905   | 0.096   | 0.905   | 0.101   | 0.901   | 0.282   | 0.709   | 0.071   | 0.944   |
> > > | 10-step | 0.146   | 0.872   | 0.153   | 0.861   | 0.159   | 0.844   | 0.347   | 0.658   | 0.119   | 0.895   |
> > >
> > > Additionally, in image generation tasks, reducing the number of inference steps in flow-matching-based text-to-image models typically results in significant quality degradation. We believe this phenomenon arises from the high complexity and variability introduced by diverse textual prompts. In contrast, language-agnostic perception tasks exclude the influence of such textual conditioning, which we believe accounts for why prior works, such as One Diffusion, require 50 to 100 inference steps to achieve satisfactory denoising results, whereas our model performs effectively with only a few inference steps, even 1 inference step.
> > >
> > > **2 Missing in the Submission**
> > >
> > > In the submitted manuscript, we only discuss the results of directly training the model to perform one-step denoising from pure noise to the target, without covering this aspect. This is because: although we have observed that the model supports few-step inference, the sensitivity to the number of inference steps varies across different tasks. As we were still in the process of thoroughly investigating this phenomenon and had not yet reached solid conclusions, we chose not to include these findings in the submission in order to maintain the academic rigor of the paper.
> > >
> > > The number of inference steps is indeed a crucial issue, which holds significant implications for efficiently leveraging multi-task diffusion models to solve perceptual tasks in the future. We sincerely apologize for this omission and will clarify it in the revised version of the manuscript.
> > >
> > > **3 VAE Decoder**
> > >
> > > The experiment is conducted with the unmodified VAE. All experimental settings are kept consistent throughout. **None of our experiments (including few-step inference) utilized any additional techniques.**
> > >
> > > -----------------
> > >
> > > We would like to once again express our gratitude for your professional and thoughtful suggestions and further comments! Any further discussion is warmly welcomed!

---

> > > > ### Comment · Reviewer_Taie · 2025-08-04
> > > >
> > > > Excellent! These are crucial points, and please remember to integrate them into the revisions. I will increase my score for the additional efforts and achievements you have made.

---

> ### Author Response · Authors · 2025-08-04
>
> We certainly will! We sincerely thank you again for your professional and constructive feedback. We are very happy and grateful for the opportunity to engage in such a meaningful discussion!

---

### Official Review · Reviewer_MKSc · 2025-06-22

**Clarity:** 4
**Significance:** 3
**Originality:** 3
**Rating:** 5
**Confidence:** 4

**Summary:**

This work proposes a DiCeption framework, which leverages the generative priors in diffusion-based pre-trained DiT model to tackle diverse visual downstream perception tasks under constrained computational resources and training samples. Through extensive experiments and ablations, this work validates several key designs to develop a visual generalist perception model without complex architectural design or large scale data.

**Questions:**

Please refer to the 'Weaknesses' section above.

**Ethical Concerns:**

["NO or VERY MINOR ethics concerns only"]

**Final Justification:**

1. The authors have shown performance on certain high-level task, that said, referring segmentation. Yet it is not representative enough, and the method's generalization on high-level tasks still need more explorations. This also aligns with Reviewer Taie's questions on the "emergent behaviors" of this method. Therefore I would not further raise my score.
2. The authors have shown results demonstrating the tradeoffs between inference time and performance gain across different tasks and datasets, along with analysis. The method achieves certain level of robustness.

**Limitations:**

Yes.

**Paper Formatting Concerns:**

None.

**Quality:**

3

**Strengths And Weaknesses:**

**Strengths**

1. It is a promising direction to tackle visual perception tasks using generative priors, and the problem is of practical importance.

2. It is non-trivial to develop a generalist model for diverse visual perception tasks, especially under limited computational resources and data samples. But this work has innovatively leveraged the powerful diffusion-based pre-trained model as a prior, and designed a simple but effective pipeline with necessary modules (such as unifying all task representations into RGB space) to transfer the generative knowledge into visual perception tasks.

3. Experimental results show the efficiency and effectiveness of this work compared with SOTA methods. This method also showcases generalizability to new data domains (_e.g._, lung and tumor) using only limited training samples (_i.e._, 50 per task).

4. The ablation analysis is thorough, which covers architectural design choices (_e.g._, UNet _vs._ DiT) as well as training/inference strategies and provides insights in how to re-purpose a diffusion-based pre-trained DiT into a generalist perception model.

**Weaknesses**

1. This framework has mainly shown its potential in low-level visual perception tasks, but it would be better to also demonstrate whether this framework can also tackle high-level visual recognition tasks, thus truly implement a **generalist image understanding model**.

2. Inference time seems to be a bottleneck due to the iterative sampling nature of diffusion model. Comparisons on inference efficiency with existing methods are suggested. In the paper, the authors have discussed the failure cases of using one-step inference. More quantitative results to show the tradeoff between sampling steps and performance are also encouraged.

---

> ### Author Rebuttal · Authors · 2025-07-30
>
> Thank you for your positive and constructive comments and insightful suggestions on our paper. We address the comments and questions as follows:
>
> **W.1 High-level Task**
>
> Thank you for your feedback. We have conducted additional experiments to evaluate our model on the RefCOCOg dataset for referring segmentation. The results demonstrate that our approach is effective for high-level visual tasks. **Unfortunately image submission in the rebuttal is not allowed. We will also include more qualitative visualizations in the revised version of the paper.**
>
> |          | RefCOCOg-val | RefCOCOg-test |
> |:--------:|:------------:|:-------------:|
> | LISA-7B  | 67.9         | 70.6          |
> | PixelLM-7B | 69.3         | 70.5          |
> | Ours     | **69.9**         | **71.2**          |
>
>
> **W.2 Inference Time**
>
> We sincerely thank you for the insightful comment.
>
> **We Naturally Support Few-step Inference**
>
> **We conduct experiments and observe that our model inherently supports few-step inference for perception tasks without any additional techniques, and shows very little performance degradation.** The effectiveness of few-step acceleration varies across different tasks. For tasks such as depth and surface normal estimation, the number of inference steps can be reduced to as few as one with acceptable slight performance degradation. For more complex tasks such as interactive segmentation, the model is still able to achieve comparable results using significantly fewer steps while maintaining competitive performance, for example, 7 step, which is still a 4x acceleration, as demonstrated below:
>
> |  Depth  | KITTI       |             | NYUv2       |             | ScanNet     |             | DIODE       |             | ETH3D       |             |
> |:--------|:------------|:------------|:------------|:------------|:------------|:------------|:------------|:------------|:------------|:------------|
> |         | absrel↓      | delta_1↑     | absrel↓      | delta_1↑     | absrel↓      | delta_1↑     | absrel↓      | delta_1↑     | absrel↓      | delta_1↑     |
> | 28-step | 0.069       | 0.949       | 0.061       | 0.960       | 0.072       | 0.944       | 0.289       | 0.722       | 0.050       | 0.975   |
> | 14-step | 0.077       | 0.942       | 0.063       | 0.958       | 0.074       | 0.943       | 0.272       | 0.718       | 0.048       | 0.978   |
> | 7-step  | 0.081       | 0.939       | 0.065       | 0.953       | 0.078       | 0.943       | 0.286       | 0.714       | 0.052       | 0.971    |
> | 3-step  | 0.083       | 0.938       | 0.069       | 0.953       | 0.077       | 0.940       | 0.294       | 0.707       | 0.063       | 0.967    |
> | 1-step  | 0.086       | 0.936       | 0.072       | 0.945       | 0.076       | 0.937       | 0.305       | 0.702       | 0.065       | 0.967    |
>
> | Normal    |         |       | nyu   |         |         |
> |:-------------|:------------|:----------|:--------|:--------|:--------|
> |              | mean ↓      | med ↓     | 11.25 ↑ | 22.5 ↑  | 30 ↑    |
> | 28-step      | 18.338      | 10.106    | 52.850  | 77.079  | 82.903  |
> | 14-step | 18.631      | 10.463    | 52.837  | 75.288  | 81.682  |
> | 7-step  | 18.335      | 10.492    | 52.771  | 75.443  | 81.936  |
> | 3-step  | 18.067      | 10.417    | 53.046  | 76.500  | 81.673  |
> | 1-step  | 18.094      | 10.382    | 51.839  | 76.575  | 81.371  |
>
> | Normal       |        |       | SCANNET |     |         |
> |:-------------|:------------|:----------|:--------|:--------|:--------|
> |              | mean ↓      | med ↓     | 11.25 ↑ | 22.5 ↑  | 30 ↑    |
> | 28-step      | 18.842      | 10.266    | 53.610  | 74.895  | 82.864  |
> | 14-step | 18.337      | 10.579    | 53.223  | 75.533  | 82.631  |
> | 7-step  | 19.008      | 10.363    | 52.628  | 74.886  | 82.055  |
> | 3-step  | 19.337      | 10.329    | 52.223  | 75.731  | 82.081  |
> | 1-step  | 19.386      | 10.395    | 52.139  | 75.492  | 81.879  |
>
> | Normal    |             |           | DIODE   |         |         |
> |:-------------|:------------|:----------|:--------|:--------|:--------|
> |              | mean ↓      | med ↓     | 11.25 ↑ | 22.5 ↑  | 30 ↑    |
> | 28-step    | 16.297      | 11.117    | 50.548  | 83.325  | 88.774  |
> | 14-step | 16.131      | 11.463    | 50.849  | 83.391  | 88.829  |
> | 7-step  | 16.835      | 11.330    | 50.039  | 82.443  | 88.218  |
> | 3-step  | 17.205      | 12.047    | 50.046  | 83.010  | 87.531  |
> | 1-step  | 17.004      | 11.849    | 49.808  | 82.972  | 87.582  |
>
> | Interactive Segmentation | 28-step  | 14-step  | 7-step | 3-step | 1-step  |
> |:-----|:-----|:-----|:------|:------|:------|
> | mIoU of 23 validation datasets | 47.10| 47.01| 46.89 | 45.18 | 42.53 |
>
> **Analysis**
>
> **We believe this is because flow matching explicitly imposes linear constraints at each intermediate denoising step—specifically, each noisy latent is constructed as a linear interpolation between the pure noise and the target signal.** This process effectively straightens the denoising trajectory, allowing the model to follow an approximately linear path even when using only a few inference steps. In contrast, if the model is trained solely with one-step denoising, the intermediate steps are not constrained and lacks this linear constraint across the trajectory, thus producing poor results as we show in Section 4.3. In contrast, traditional ODE-based diffusion models do not impose such linear trajectory constraints, and therefore cannot support inference with few denoising steps (such as 4 steps) after being trained with multi-step denoising (such as 50 steps).
>
> Our additional experiment proves this. We further experiment with PixArt-α, which uses a DiT-style architecture but adopts a standard ODE-based scheduler. Its results significantly deteriorate when the number of inference steps is reduced, further supporting our analysis above.
>
> |         | KITTI   |         | NYUv2   |         | ScanNet |         | DIODE   |         | ETH3D   |         |
> |:--------|:--------|:--------|:--------|:--------|:--------|:--------|:--------|:--------|:--------|:--------|
> |         | absrel↓  | delta_1↑ | absrel↓  | delta_1↑ | absrel↓  | delta_1↑ | absrel↓  | delta_1↑ | absrel↓  | delta_1↑ |
> | 20-step | 0.093   | 0.905   | 0.096   | 0.905   | 0.101   | 0.901   | 0.282   | 0.709   | 0.071   | 0.944   |
> | 10-step | 0.146   | 0.872   | 0.153   | 0.861   | 0.159   | 0.844   | 0.347   | 0.658   | 0.119   | 0.895   |
>
> In image generation tasks, simply reducing inference steps in a flow-matching-based text-to-image model also leads to noticeable quality degradation. This is due to the high complexity and variability introduced by diverse text prompts. In contrast, our perception tasks eliminate the influence of textual prompts, which we believe explains why prior works like One Diffusion require 50~100 inference steps for denoising while ours works well with just a few steps.
>
> For comparisons on inference efficiency, we select One Diffusion as baseline and conduct a comparative study on our shared task—depth estimation—under varying numbers of inference steps. Unlike One Diffusion, which suffers from significant performance degradation during few-step inference and fails to produce reasonable results in the 1-step setting, our method is capable of generating high-quality outputs even with just a single inference step. The results demonstrate that our method significantly outperforms One Diffusion in both efficiency and output quality.
>
> |                           | KITTI   |         | NYUv2   |         | ScanNet |         | DIODE   |         | ETH3D   |         |
> |:--------------------------|:--------|:--------|:--------|:--------|:--------|:--------|:--------|:--------|:--------|:--------|
> |                           | absrel↓  | delta_1↑ | absrel↓  | delta_1↑ | absrel↓  | delta_1↑ | absrel↓  | delta_1↑ | absrel↓  | delta_1↑ |
> | Ours-28-step (default setting, inference time 2s) | 0.069   | 0.949   | 0.061   | 0.960   | 0.072   | 0.944   | 0.289   | 0.722   | 0.050   | 0.975   |
> | Ours-7-step (reduce to 1/4) | 0.081   | 0.939   | 0.065   | 0.953   | 0.078   | 0.943   | 0.286   | 0.714   | 0.052   | 0.971   |
> | Ours-1-step               | 0.086   | 0.936   | 0.072   | 0.945   | 0.076   | 0.937   | 0.305   | 0.702   | 0.065   | 0.967   |
> | OD-50step (default setting, inference time 6s) | 0.101   | 0.908   | 0.087   | 0.924   | 0.094   | 0.906   | 0.399   | 0.661   | 0.072   | 0.949   |
> | OD-12step (reduce to 1/4) | 0.142   | 0.867   | 0.114   | 0.871   | 0.128   | 0.853   | 0.411   | 0.659   | 0.092   | 0.910   |
> | OD-1step                  | FAIL    | FAIL    | FAIL    | FAIL    | FAIL    | FAIL    | FAIL    | FAIL    | FAIL    | FAIL    |
>
> **Revise the according parts in Section 4.3**
>
> We apologize for the confusing presentation in Section 4.3. **In Section 4.3, we say one-step inference is ineffective is because those results are obtained with the model trained to predict the target directly from noise in a single forward pass, which follows the setting of previous approach Genpercept.** Here we show that when the model is trained under a multi-step flow matching denoising setting, it can naturally support few-step inference. In the revised version of the paper, we will clarify this point and provide a more complete analysis by incorporating additional results that demonstrate the few-step inference capability enabled by multi-step flow matching training.
>
> Once again, we sincerely express our sincere gratitude for your insightful and professional comments! Please let us know if you have any further concerns.

---

> > ### Comment · Reviewer_MKSc · 2025-08-02
> >
> > Thanks for the author's response. It has addressed my concerns and hence I would keep my score.

---

> > > ### Author Response · Authors · 2025-08-04
> > >
> > > We sincerely thank you again for your valuable comments, insightful suggestions, and constructive discussion. :)

---

### Official Review · Reviewer_1bZo · 2025-06-30

**Clarity:** 3
**Significance:** 2
**Originality:** 2
**Rating:** 4
**Confidence:** 3

**Summary:**

The authors propose a multi-task perception model that addresses various image understanding tasks within a unified framework by leveraging the large-scale prior knowledge of a pre-trained text-to-image (TTI) model.

Specifically, they fine-tune the pre-trained Stable Diffusion 3 (SD3) model to accept three types of inputs: the image to be analyzed, a task prompt that specifies the target domain for generation, and selective point embeddings that guide point-prompted interactive segmentation. These inputs are concatenated token-wise, enabling the model to generate target-domain images that serve as the understanding results.

By simultaneously training the model on a large, unified dataset encompassing diverse image understanding tasks and building upon the prior knowledge of the pre-trained TTI model, the proposed approach demonstrates superior performance compared to existing multi-task baselines. Furthermore, the paper presents comprehensive experimental results and ablation studies to validate the effectiveness of the proposed method.

**Questions:**

**1. Comparison to One Diffusion**

In L139, you state that One Diffusion requires detailed prompts to achieve better results. If so, why not generate such detailed prompts using an image captioning model instead of relying on human-provided prompts? My understanding (which may not be entirely correct) is that perception tasks should ideally not depend on subjective human interpretation, as detailed prompts from humans already incorporate human-level understanding. Using prompts generated by a pre-trained image captioning model could mitigate this dependency while still enhancing performance.

**2.Ambiguous experimental settings and terminology**

The experimental setup described in Section B.5 regarding few-shot fine-tuning of SD3 is unclear, especially considering that Deception itself also fine-tunes a pre-trained SD3 model. What is the fundamental difference between these two approaches?
Additionally, the concept of "pixel-level alignment" mentioned in Section B.6 is not introduced or explained in the main paper, making it difficult to understand its role or significance.

**3.Unclear motivation for CFG**

The exact motivation for incorporating Classifier-Free Guidance (CFG) into the proposed method is not clearly articulated, making it difficult to understand its intended role and benefits.

**Ethical Concerns:**

["NO or VERY MINOR ethics concerns only"]

**Final Justification:**

While I had some initial concerns, the authors’ rebuttal addressed them effectively. In light of the reviews and discussions with other reviewers, I have come to recognize the merits of the paper and have accordingly increased my score.

**Limitations:**

Yes

**Paper Formatting Concerns:**

There is no formatting issues

**Quality:**

2

**Strengths And Weaknesses:**

# Strengths

**1. Simple yet effective approach for a unified multi-task perception model**

The proposed method employs a straightforward token-wise concatenation of image conditions and task prompts, followed by fine-tuning with a pre-training flow matching loss. Despite its simplicity, the method achieves superior performance compared to existing multi-task baselines.

**2. Leveraging prior knowledge from pre-trained TTI models**

By utilizing the prior knowledge embedded in pre-trained text-to-image (TTI) models, the training process becomes more efficient and effective than approaches that require training entirely from scratch.

# Weaknesses

**1. Incremental improvement over One Diffusion**

The primary distinction from One Diffusion lies in the use of pre-trained models as the initialization point for training (+ some miscellaneous techniques). While pre-trained models indeed simplify and stabilize the training process, the proposed method appears to be only an incremental extension of One Diffusion, even considering the introduction of point embeddings, which are claimed to be novel in this work.

Furthermore, One Diffusion was trained from scratch using both image understanding and image generation tasks. In contrast, the proposed approach leverages a TTI model pre-trained for text-to-image generation, followed by fine-tuning for image understanding tasks. Given this difference, a direct comparison in terms of data requirements may not be entirely fair.

**2. Substantial computational requirements**

Despite leveraging a pre-trained model, the proposed Deception method still requires large-scale training resources. Considering these substantial computational demands, it remains unclear to what extent the advantages of using a pre-trained model are truly realized. More thorough ablation studies are needed to isolate and validate the impact of pre-training.

**3. Limited experimental ablations**

Although the paper claims that the proposed method can adapt to novel tasks with as few as 50 images, this is not adequately supported by experiments. Only qualitative results for few-shot training are provided, without quantitative comparisons against baselines, making it difficult to assess the true effectiveness of few-shot adaptation.

---

> ### Author Rebuttal · Authors · 2025-07-30
>
> We sincerely thank the reviewer for the constructive feedback.
>
> ### **Weaknesses**
> **W.1 Incremental improvement over One Diffusion**
>
> We respectfully disagree with the claim that our work is merely an incremental improvement over One Diffusion.
> The most fundamental difference lies in the target we want to address. We focus only on perception tasks rather than generation tasks in One Diffusion, which significantly leads to different research goals and conclusions. Specifically, our valuable findings and conclusions, such as suitable diffusion architectures for multiple perception tasks, input strategies, role of CFG, **and our inherent support for few-step inference** are all towards better repurposing diffusion models to a generalist perception model. Especially our method's direct support for few-step inference shows a salient distinction to One Diffusion.
>
> **Key Distinction and Innovation: Enabling Few-Step Inference for Multi-Task Perception with Diffusion Models**
>
> To further highlight the advantages and contributions of our approach compared to One Diffusion, we investigate the feasibility of few-step, even 1-step inference of our method. Our findings reveal that the proposed model achieves high-quality results under significantly reduced inference steps, with minimal performance degradation. **To the best of our knowledge, this is the first time such a capability is demonstrated in a diffusion-based multi-task perception model, providing strong evidence for the effectiveness in efficient inference for perceptual tasks. This highlights our contributions.**
>
> We believe the superior few-step inference capability of our model stems from the intrinsic properties of flow matching, which enforces linear constraints at every intermediate denoising step. In particular, each noisy latent is explicitly constructed as a linear interpolation between pure noise and the target signal. This linearity effectively regularizes the denoising trajectory, encouraging the model to follow a near-linear path in latent space, which remains stable even under a reduced number of inference steps.
>
> **We also apologize for the unclear explanation in Section 4.3.** The failure of one-step inference reported in that section stems from the fact that the model was trained to predict the target directly from pure noise in a single step, without involving intermediate latent states, following the strategy of GenPercept. In contrast, here we demonstrate that when the model is trained using a multi-step denoising process with intermediate noisy latents, it can naturally support few-step inference without additional techniques. We will revise the paper for better clarity.
>
> In contrast, One Diffusion requires 50 to 100 inference steps to produce satisfactory results, while our model maintains high performance with significantly fewer steps. We attribute this discrepancy to One Diffusion’s strong reliance on detailed text prompts, which introduces substantial variability into the denoising process and increases the complexity of the underlying data distribution that the model must learn. Similarly, in flow-matching-based text-to-image generation models, aggressively reducing the number of inference steps often leads to notable quality degradation. These observations both suggest that the presence of complex textual prompts complicates the generative process, necessitating more denoising steps. This, in turn, supports the design choice in our work to eliminate complex textual conditioning in perception tasks, thereby simplifying the target distribution and enabling efficient and effective few-step inference.
>
> |  Depth | KITTI   |         | NYUv2   |         | ScanNet |         | DIODE   |         | ETH3D   |         |
> |:---|:--------|:--------|:--------|:--------|:--------|:--------|:--------|:--------|:--------|:--------|
> |     | absrel↓  | delta_1↑ | absrel↓  | delta_1↑ | absrel↓  | delta_1↑ | absrel↓  | delta_1↑ | absrel↓  | delta_1↑ |
> | Ours-28-step (default setting) | 0.069   | 0.949   | 0.061   | 0.960   | 0.072   | 0.944   | 0.289   | 0.722   | 0.050   | 0.975   |
> | Ours-7-step (reduce to 1/4) | 0.081   | 0.939   | 0.065   | 0.953   | 0.078   | 0.943   | 0.286   | 0.714   | 0.052   | 0.971   |
> | Ours-1-step    | 0.086   | 0.936   | 0.072   | 0.945   | 0.076   | 0.937   | 0.305   | 0.702   | 0.065   | 0.967   |
> | OD-50step (default setting) | 0.101   | 0.908   | 0.087   | 0.924   | 0.094   | 0.906   | 0.399   | 0.661   | 0.072   | 0.949   |
> | OD-12step (reduce to 1/4) | 0.142   | 0.867   | 0.114   | 0.871   | 0.128   | 0.853   | 0.411   | 0.659   | 0.092   | 0.910   |
> | OD-1step  | FAIL | FAIL| FAIL| FAIL | FAIL| FAIL| FAIL| FAIL| FAIL| FAIL|
>
> We provide additional few-step inference results across different tasks. Our method effectively reduces the number of inference steps while incurring only minimal performance degradation. For complex tasks such as interactive segmentation, although 1-step inference remains infeasible, our method reduces the required steps to 7, achieving a 4× speed-up compared to the original setting while preserving strong performance.
>
> | Normal    |    |   | nyu   |   | |
> |:----|:---|:--|:-----|:----|:----|
> |              | mean ↓      | med ↓     | 11.25 ↑ | 22.5 ↑  | 30 ↑    |
> | 28-step      | 18.338      | 10.106    | 52.850  | 77.079  | 82.903  |
> | 14-step | 18.631 | 10.463    | 52.837  | 75.288  | 81.682  |
> | 7-step  | 18.335 | 10.492    | 52.771  | 75.443  | 81.936  |
> | 3-step  | 18.067 | 10.417    | 53.046  | 76.500  | 81.673  |
> | 1-step  | 18.094 | 10.382    | 51.839  | 76.575  | 81.371  |
>
> | Normal       |    | | SCANNET |  | |
> |:-----|:-----|:----|:--|:--|:----|
> |   | mean ↓      | med ↓     | 11.25 ↑ | 22.5 ↑  | 30 ↑    |
> | 28-step      | 18.842      | 10.266    | 53.610  | 74.895  | 82.864  |
> | 14-step | 18.337  | 10.579    | 53.223  | 75.533  | 82.631  |
> | 7-step  | 19.008  | 10.363    | 52.628  | 74.886  | 82.055  |
> | 3-step  | 19.337  | 10.329    | 52.223  | 75.731  | 82.081  |
> | 1-step  | 19.386  | 10.395    | 52.139  | 75.492  | 81.879  |
>
> | Interactive Segmentation | 28-step  | 14-step  | 7-step | 3-step | 1-step  |
> |:-----|:-----|:-----|:---|:-----|:---|
> | mIoU of 23 validation datasets | 47.10| 47.01| 46.89 | 45.18 | 42.53 |
>
> In summary, **as acknowledged by other reviewers, our work goes well beyond simple incrementation.** It provides a thorough and systematic exploration of diffusion models in the context of multi-task visual perception, establishing both novel capabilities and practical advantages.
>
> **W.2 On computational cost**
>
> We appreciate the concern regarding computational overhead. However, we would like to clarify that our training cost is acceptable compared to existing efforts:
> - SAM 256 A100 GPUs
> - One Diffusion 64 H100 GPUs
> - ours 4 H100 GPUs
>
> We believe this makes our setup significantly more accessible and reproducible for the community.
> As for inference efficiency, we further show that our model supports few-step inference without significant degradation in performance, substantially reducing runtime cost at test time. Furthermore, we will release our model to facilitate future research and development.
>
>
> **W.3 Quantitative Evaluation on Few-shot Training**
>
> Thank you for the valuable suggestion. We have applied the same set of 50 images for fine-tuning across ours and baseline methods. The results demonstrate the effectiveness of our approach in few-shot scenarios.
>
> | mIoU  | ours-50img | MedicalSAM2-50img | MedSegDiff-50img |
> |:---|:---:|:----:|:----:|
> | Brain Tumor | 0.942  | 0.863 | 0.837 |
> | Lung | 0.933 | 0.885    | 0.826  |
>
> |   | SSIM  | PSNR  |
> |:-----|:------|:------|
> | Ours-50img  | 0.920 | 27.32 |
> | Painter-50img | 0.846 | 20.53 |
> | MIRNet-v2-50img | 0.857 | 22.84 |
>
> ### **Questions**
>
> **Q.1 On Prompt Generation Using Image Captioning Models**
>
> We acknowledge the use of image captioning models as a possible solution of task prompts. However, these captions:
> - Ultimately still originate from human knowledge.
> - Different models and hyperparameters may yield different prompts for the same image.
> - Captioning models have **hallucination problems**, a well-known and unresolved issue.
> - Generating high-quality captions typically requires large models, which introduces significant **additional computational cost.**
> - Fail to reduce inference step as we show in W.1. Even more steps (50-100) are required.
>
> Therefore, relying on caption-generated prompts does not fundamentally solve the problem. For language-agnostic perception tasks, we argue that introducing unnecessary language components is neither essential nor clearly beneficial.
> Additionally, as we mentioned, our method demonstrates strong few-step inference capabilities, whereas One Diffusion sometimes requires up to 100 steps to produce results. We believe this is due to the complex prompts it needs, which necessitates more timesteps to arrive at a reasonable prediction.
>
> **Q.2 Clarification on Adaptation Experiment and "Pixel-aligned Training"**
>
> We apologize for the lack of clarity. This experiment involves using SD3 and Diception as base models, and applying LoRA-based fine-tuning on just 50 training images to demonstrate task adaptation.
>
> The term pixel-aligned training refers to training on pixel-aligned perception tasks (e.g., depth, normal, segmentation, pose). After being trained on perceptual tasks, our model demonstrates faster and more effective adaptation to new tasks than SD3. We will clarify this point in the revised manuscript to avoid any potential ambiguity.
>
> **Q.3 Motivation for CFG**
>
> CFG is a widely used technique in conditional diffusion models for improving image generation quality without additional training. Our motivation is to investigate whether such generation-oriented techniques could also enhance performance when repurposing diffusion models for perception tasks.
>
> We sincerely thank you again for your constructive feedback! Please let us know if you have any further concerns.

---

> > ### Comment · Area_Chair_PWqV · 2025-08-01
> > **Please respond**
> >
> > Dear reviewer,
> >
> > seems like your comments have been addressed. Can you please read the rebuttal and respond to authors?

---

> ### Comment · Reviewer_1bZo · 2025-08-04
>
> Thank you for your constructive response. Some of my concerns have been adequately addressed.
>
> However, the main contribution still appears to be the construction of a framework that applies techniques commonly used in generative models to diffusion models—techniques that have not been explicitly tailored for perception tasks.
>
> Moreover, the current writing of the main paper does not clearly convey the rebuttal’s claim regarding the advantage of few-step inference.

---

> > ### Author Response · Authors · 2025-08-04
> >
> > We thank the reviewer for the insightful feedback.
> >
> > First, we believe that in the domain of diffusion models for perception, valuable contributions are not limited only to the invention of new techniques. Clarifying which existing options are more effective, uncovering their performance limits, and understanding the reasons behind their success are equally important. Recent works[1,2], although based on techniques commonly used in generative diffusion models, have made significant and widely recognized contributions.
> >
> > [1] **(CVPR 2024 Oral, Best Paper Award Candidate)** Repurposing Diffusion-Based Image Generators for Monocular Depth Estimation
> >
> > [2] **(ICLR 2025)** What Matters When Repurposing Diffusion Models for General Dense Perception Tasks?
> >
> > In diffusion models for multi-task perception, our findings address a number of open questions that, to our knowledge, have not been conclusively answered before. We believe our core contribution and key novelty, lies in the systematic exploration of, and provision of solid answers to, several fundamental questions regarding the application of diffusion models to multi-task perception:
> >
> > - **Research Goal:** Our objective is to build a simple, **fully parameter-sharing** generalist model for perception tasks that does not require per-task decoders. Leveraging the strong RGB priors of diffusion models and the RGB unification of perception tasks, we adopt a diffusion-based approach.
> >
> > - **Model Architecture:** No prior work has conducted a detailed exploration of which prevalent diffusion model architecture—U-Net or DiT—is more suitable for a generalist perception model. We demonstrate that the DiT architecture holds greater potential for multi-task solutions.
> >
> > - **Optimal Input Method:** We investigated the most effective way to introduce input images when applying diffusion models to perception tasks, demonstrating that token-wise concatenation is more suitable compared to traditional channel-wise concatenation used in many prior works.
> >
> > - **Addressing Complex Tasks:** We successfully demonstrate our generalist model can solve complex tasks like interactive segmentation, which received little attention especially among multi-task models like One Diffusion.
> >
> > - **Leveraging Priors for Data Efficiency:** We show that our method can effectively leverage diffusion priors. Consequently, we achieve on-par results using significantly less data than is typically required for single-task expert models.
> >
> > - **Performance Gap:** We provide a strong argument that, within our data setting, our generalist model exhibit no noticeable performance gap compared to its single-task counterparts.
> >
> > - **Applicability of Generation Techniques to Perception:** We provide the first systematic analysis of whether techniques effective for generative tasks are also beneficial for perception. We show that CFG can be effective for certain tasks, while other prominent techniques in generation like ControlNet are not well-suited for perception.
> >
> > - **Feasibility of Few-step Inference:** We provide a thorough analysis of few-step inference for perception tasks, convincingly demonstrating the significant advantages of flow matching for perception.
> >
> > - **Advantages over Generative Diffusion Models:** We demonstrate that, unlike diffusion models primarily designed for generation, our generalist perception model is significantly more efficient for downstream perception tasks and exhibits superior detail preservation.
> >
> > In summary, the novelty of our work does not stem from introducing a single new module, but from a comprehensive set of well-motivated experiments that, for the first time, construct a fully parameter-sharing diffusion-based generalist model for perception. Through rigorous empirical studies, we address the central question of how to effectively repurpose powerful diffusion models for multiple perception tasks and provide a foundation of best practices for utilizing powerful diffusion model as a generalist perception model. Notably, other reviewers have also recognized the novelty of our work.
> >
> > **Oversight of not Including Related Discussion**
> >
> > We acknowledge and sincerely apologize for the oversight of not including related discussion in the manuscript. This is because, although we had observed that our model exhibits promising few-step inference capabilities, the sensitivity to the number of inference steps varies across different tasks. As we were still in the process of thoroughly investigating this phenomenon and we had not yet drawn robust conclusions for that time, we chose not to include these preliminary observations to uphold the rigor and clarity of the paper.
> >
> > We acknowledge that the number of inference steps is a critical factor that should be discussed, with important implications for the efficient deployment of multi-task diffusion models in perceptual tasks. We sincerely apologize for this oversight and will make sure to clarify it in the revised version of the manuscript.

---

> ### Comment · Reviewer_1bZo · 2025-08-04
>
> Thank you very much for your instantaneous response. I believe all of my concerns have been thoroughly addressed.

---

> > ### Author Response · Authors · 2025-08-04
> >
> > Thank you very much for your professional suggestions and valuable comments.

---

### Official Review · Reviewer_g92U · 2025-07-01

**Clarity:** 3
**Significance:** 3
**Originality:** 2
**Rating:** 5
**Confidence:** 3

**Summary:**

This work proposes DICEPTION, a general-purpose diffusion model designed for multiple visual perception tasks. Building upon pre-trained text-to-image diffusion models, it unifies various tasks within the RGB space. By preserving pre-trained priors through token-wise concatenation and fine-tuning, DICEPTION achieves performance comparable to state-of-the-art single-task models while utilizing merely 0.06% of their data. The paper validates the model's robustness and effectiveness through experiments conducted across diverse datasets and tasks.

**Questions:**

1. The model relies on post-processing methods for various tasks, but are there optimization schemes or alternative solutions for the errors introduced by post-processing?
2. The analysis of Classifier-free Guidance only involves depth and normal tasks. Is it applicable to segmentation tasks?
The model proposed in the article has significant advantages in efficiency. If the questions in the weaknesses and questions can be effectively explained and illustrated, I am willing to further improve my score.

**Ethical Concerns:**

["NO or VERY MINOR ethics concerns only"]

**Final Justification:**

While certain parts of this manuscript initially confused me, the authors' rebuttal has effectively addressed these issues. After reviewing the comments from other reviewers and the authors' responses, I believe this work is of significant value in the research field and is supported by adequate experiments. Therefore, I have accordingly increased my score.

**Limitations:**

The article provides a comprehensive analysis of the limitations of existing models and areas for improvement in the future.

**Paper Formatting Concerns:**

1. There are many sentences in the article that use italics. If emphasis is needed, bolding is sufficient, but the italicized content is slightly too much, which makes the overall appearance of the article not very attractive.

**Quality:**

3

**Strengths And Weaknesses:**

Strengths:
1. The model's DiT and other architectures fully preserve and leverage the prior knowledge of pre-trained models, effectively reducing training costs.
2. The paper conducts extensive and thorough experiments, verifying the model's generalization ability through comparisons across multiple scenarios, and fully analyzing the correlation between metric results and model settings.
3. The model enables effective and rapid adaptation to new task scenarios through LoRA fine-tuning technology.

Weaknesses:
1. This work emphasizes that pixel-aligned training can enhance the ability to preserve fine details, but it fails to clarify the specific implementation method of pixel alignment. For example, is this strategy applied to impose constraints at each denoising step?
2. This work states that the performance difference between the multi-task model and the single-task model is not significant, and the advantages of the model setup have not been highlighted through result verification on a larger dataset.
3. The paper places a considerable number of ablation study results on the validity of model settings in the appendix. However, these comparative results are highly relevant to verifying the effectiveness of the model's key innovations, and these model architectures also constitute the main contributions of the paper. It might be advisable to allocate some space in the related work section to the appendix, so as to better highlight the validity of the model settings in the main text.

---

> ### Author Rebuttal · Authors · 2025-07-30
>
> We sincerely thank the reviewer for the valuable feedback. We address the comments and questions as follows:
>
> ### **Weaknesses**
> **W.1 Clarification on Pixel-Aligned Training**
>
> We apologize for the confusion. In our paper, **the term "pixel-aligned training" refers to our training on pixel-level perceptual tasks demonstrated in our paper (such as depth, surface normal, segmentation, human pose).**  No additional techniques and strategies are applied. Training on these perceptual tasks effectively facilitates detail preservation when adapt to downstream tasks such as image highlighting. We will clarify this in the revised manuscript to avoid ambiguity.
>
> **W.2 Scalability on Larger Datasets**
>
> Thank you for your valuable comments. We acknowledge the limitation of not validating the scalability of our model on larger datasets. This is primarily due to constraints in computational resources. Nevertheless, the current results are still very promising and demonstrate the potential of our approach. This is supported by our comprehensive evaluation across a total of 33 widely-used benchmark datasets. We are committed to explore scalability further with more computational resources and larger training dataset in future work.
>
> **W.3 Paper Organization**
>
> Thank you for pointing this out. In the revised version, we will move some of the related work section to the appendix and highlight our experiment results and analyses in the main paper.
>
> ### **Questions**
> **Q.1 Optimization Schemes and Alternative Solutions for the Errors Introduced by Post-processing**
>
> Thank you for your constructive comments. In our paper, we have achieved quantitative metrics with a simple post-processing step, which represents a qualitative leap over prior approaches that relied solely on visualization. We regard evaluating these tasks as a fundamental challenge that must be addressed. However, prior works largely overlook this problem. As diffusion models for perception continue to evolve, relying solely on tasks such as depth and normal estimation offers only a limited view and fails to adequately evaluate the multi-task diffusion models. While our current evaluation metrics may look underperformed to traditional task-specific approaches due to the inherent errors introduced by post-processing, we believe it still provides a preliminary baseline for future diffusion-based perceptual methods.
>
> Given the considerable variation in representations across tasks, especially in instance-level settings such as pose estimation, existing methods commonly resort to individually designed decoders tailored to each task. In the context of Diffusion Model, we believe a suitable solution may lie in further exploration of optimizing the VAE decoder. Recent works [1] have shown that fine-tuning the VAE can achieve high-quality point map estimation. Inspired by these efforts, we believe optimizing VAE for decoding the final mask, and even decoding results across multiple tasks, is promising. We will delve into this problem in detail in our future work.
>
> [1] GeometryCrafter: Consistent Geometry Estimation for Open-world Videos with Diffusion Priors
>
> Additionally, we also provide the results of some simple alternative solutions:
>   - For pose estimation, we conduct a primary experiment on a more complex heatmap-based keypoint representation.
>
> |         | HRNet | HRFormer | ViTPose | Painter | Ours | Ours-Heatmap |
> |:-------:|:-----:|:--------:|:-------:|:-------:|:----:|:------------:|
> | **AP** | 76.3  | 77.2     | 78.3    | 72.5    | 57.8 | 68.9         |
>
>   - For segmentation, we randomly sample points within each RGB mask and re-run the interactive segmentation. This can lead to improvement, but does not address the root problem.
>
> |    | SparK | OneFormer | Mask2Former | Ours | Ours-point |
> |:--:|:-----:|:---------:|:-----------:|:----:|:----------:|
> | **AP** | 45.1  | 49.2      | 50.1        | 33.2 | 41.7       |
>
> **These result shows our method could benefit from improved post-processing.** We believe the result still has room for improvement. However, to tackle all instance level perception tasks, we assume the RGB space is good for visualizations, but not efficient as specially designed decoders, which leads to future research questions. In summary, to fundamentally address this problem, we believe the exploration of a general multi-task VAE decoder holds potential.
>
> **Q.2 CFG for segmentation**
>
> Thank you for pointing this out. We conduct additional experiments and show that CFG has negligible impact on segmentation quality, which is consistent with the visualizations presented in Figure S5 of the supplementary material.
>
> |      | cfg=1 | cfg=2 | cfg=3 | cfg=4 | cfg=5 |
> |:----:|:-----:|:-----:|:-----:|:-----:|:-----:|
> | mIoU on 23 validation sets | 47.10 | 47.12 | 47.08 | 46.91 | 46.57 |
>
>
> **Paper Formatting Concerns**
>
> Thank you for the suggestion. We appreciate the feedback and will revise the manuscript to reduce the use of italics and apply emphasis more appropriately to improve readability and presentation.
>
>
>
> We sincerely thank you again for your highly valuable and constructive suggestions! Please let us know if you have any further concerns.

---

> ### Comment · Reviewer_g92U · 2025-08-05
>
> Thanks to the authors for their thorough responses to my review comments. The major concerns have been addressed. Therefore, I will upgrade the score to "accept". Hope this job could open up future research.

---

> > ### Author Response · Authors · 2025-08-05
> >
> > We sincerely thank you again for your time, constructive feedback, and valuable suggestions!

---

### Note · Authors · 2025-08-12

We sincerely thank the AC and all reviewers for their time and effort throughout the entire review process. We are very encouraged that reviewers recognize the importance of our research and appreciate the value of our contribution.

We thank all the professional reviewers for their insightful and valuable comments, and we truly appreciate such a valuable opportunity to engage in the constructive discussions with all responsible reviewers. We have carefully studied and incorporated all the feedback and have substantially revised our manuscript to address the main concerns raised.

During the rebuttal, we provided extensive additional experiments, including referring segmentation, a quantitative evaluation of few-shot fine-tuning, potential improvements to post-processing, and support for few-step inference. We also addressed the writing issues pointed out by the reviewers and engaged in discussions on computational cost and emergent capabilities. We are pleased to have thoroughly and convincingly resolved all concerns and are pleased to see the reviewers' positive rating and feedback.

The final version of our manuscript will be thoroughly revised to incorporate the valuable suggestions garnered during the rebuttal process, along with our new experimental results, to significantly enhance its overall quality. We believe our work makes a significant contribution to the development of diffusion-based unified perception models. Once again, we truly appreciate the dedication of AC and all reviewers, insightful and valuable comments, and the opportunity to further strengthen our work.

---

### Decision · Program_Chairs · 2025-09-17

**Decision:**

Accept (spotlight)

**Comment:**

This paper presents a unified diffusion-based approach to visual perception, tackling a broad range of tasks through a single framework. While reviewers initially raised several important concerns—including clarity, generalization to high-level tasks, and computational trade-offs—the authors thoroughly addressed these in the rebuttal with substantial new experiments and detailed analysis. Three reviewers increased their scores following the discussion, and the overall consensus is that the paper makes a meaningful and timely contribution to the field. The Area Chair concurs with this consensus and recommends accepting the paper. The authors are encouraged to incorporate the additional experiments and revisions into the final version to further strengthen the work.